# The complex of ferric-enterobactin with its transporter from *Pseudomonas aeruginosa* suggests a two-site model

Lucile Moynié [1,2,3,10], Stefan Milenkovic [4,10], Gaëtan L.A. Mislin [5,6,10], Véronique Gasser [5,6], Giuliano Malloci [4], Etienne Baco[5,6], Rory P. McCaughan [7], Malcolm G.P. Page [8], Isabelle J. Schalk[5,9], Matteo Ceccarelli [4,9] & James H. Naismith [1,2,3]

Bacteria use small molecules called siderophores to scavenge iron. Siderophore-$Fe^{3+}$ complexes are recognised by outer-membrane transporters and imported into the periplasm in a process dependent on the inner-membrane protein TonB. The siderophore enterobactin is secreted by members of the family Enterobacteriaceae, but many other bacteria including *Pseudomonas* species can use it. Here, we show that the *Pseudomonas* transporter PfeA recognises enterobactin using extracellular loops distant from the pore. The relevance of this site is supported by in vivo and in vitro analyses. We suggest there is a second binding site deeper inside the structure and propose that correlated changes in hydrogen bonds link binding-induced structural re-arrangements to the structural adjustment of the periplasmic TonB-binding motif.

[1] Division of Structural Biology, Wellcome Trust Centre of Human Genomics, 7 Roosevelt Drive, Oxford OX3 7BN, UK. [2] The Research Complex at Harwell, Harwell Campus, Oxfordshire OX11 0FA, UK. [3] The Rosalind Franklin Institute, Didcot OX11 0FA, UK. [4] Department of Physics, University of Cagliari, Cittadella Universitaria, SP Monserrato-Sestu Km 0.700, Monserrato 09042, Italy. [5] Université de Strasbourg, UMR7242, ESBS, Bld Sébastien Brant, Illkirch F-67413 Strasbourg, France. [6] CNRS, UMR7242, ESBS, Bld Sébastien Brant, Illkirch F-67413 Strasbourg, France. [7] BSRC, The University of St Andrews, St Andrews KY16 9ST, UK. [8] Department of Life Sciences & Chemistry, Campus Ring 1, Bremen 28759, Germany. [9] Istituto Officina dei Materiali-CNR, Cittadella Universitaria, Monserrato 09042, Italy. [10] These authors contributed equally: Lucile Moynié, Stefan Milenkovic, Gaëtan L. A. Mislin. Correspondence and requests for materials should be addressed to I.J.S. (email: schalk@unistra.fr) or to M.C. (email: matteo.ceccarelli@dsf.unica.it) or to J.H.N. (email: naismith@strubi.ox.ac.uk)

Bacteria, like all other forms of life on earth, require the transition metal iron for growth, thus acquisition of iron from the environment is essential. The very low aqueous solubility of Fe(III) oxide, the dominant form in the oxygen-rich aqueous environment of earth, rules out reliance on passive diffusion for uptake. Bacteria have evolved an active-uptake strategy in which they synthesise low-molecular-weight molecules, known as siderophores, that chelate $Fe^{3+}$ ions with a very high affinity, forming soluble complexes[1]. In Gram-negative bacteria, $Fe^{3+}$-siderophore complexes are recognised at the bacterial cell surface by highly selective outer-membrane TonB-dependent transporters (TBDTs)[2]. These transporters bind the $Fe^{3+}$-siderophore complex and translocate it into the periplasm. The energy necessary for this uptake mechanism is provided by the cytoplasmic membrane potential and mediated by the inner membrane energy-transducing TonB-ExbB-ExbD protein complex interacting with a small periplasmic N-terminal region of the transporter called "TonB box"[3]. Once the ferric-siderophore has been transported across the outer membrane, iron will be released either into the periplasm or into cytoplasm, depending on the system. The latter scenario requires inner-membrane proteins, either ATP-binding cassette transporters or proton-motive-force-dependent transporters[4]. TBDTs have a characteristic two-domain organisation with a transmembrane C-terminal β-barrel and a globular N-terminal plug domain located inside the barrel[5]. Current models of translocation envisage binding of the $Fe^{3+}$-siderophore, triggering engagement with TonB which results in a change in the conformation of the plug domain to create a channel permitting transit of the $Fe^{3+}$-siderophore[5–8].

Enterobactin is an archetypical siderophore with exceptionally high affinity for $Fe^{3+}$ ($K_a = 10^{52}$)[9]. Many Gram-negative bacteria, including *E. coli*[10], produce and take up this siderophore. Other bacteria do not synthesise the molecule themselves, but will use it to scavenge iron in a piracy strategy[11]. Enterobactin comprises three catecholamide groups linked by a trilactone ring, and when not bound to iron it is easily degraded[9,12] (Fig. 1a). The $Fe^{3+}$ is coordinated in an octahedral manner by the six hydroxyls of the three catecholates, which have been reported to be deprotonated at neutral pH[9] resulting in a net charge for the molecule of −3. In *E. coli*, $Fe^{3+}$-enterobactin is recognised at the bacterial cell surface and transported into the periplasm by FepA, a TBDT. Previous studies have suggested that the $Fe^{3+}$-enterobactin recognition by FepA, is a multi-step process involving an initial binding event followed by a secondary binding event that proceeds to translocation[13]. The structure of FepA has been solved[14], and although the authors located one, possibly two, iron atoms (by anomalous scattering) they were unable to model either the siderophore or its interactions with the protein. To date, no structure of enterobactin bound to its TBDT has been reported. Site-directed mutagenesis and loops deletion identified several basic and aromatic residues on FepA that when mutated affected the binding or the uptake of the siderophore[15–18]. The structures of pyoverdine, pyochelin, citrate and ferrichrome iron complexes bound to their respective transporters FpvA, FptA, FecA and FhuA showed that the ferric-siderophore complexes were bound inside the barrel adjacent to the plug domain, a position thought to be poised for translocation[5,19–23]. *Pseudomonas aeruginosa*, a serious opportunistic pathogen, does not make enterobactin but does possess a TBTD outer-membrane protein PfeA, related to FepA, that binds and translocates $Fe^{3+}$-enterobactin into the periplasm[24].

With the emergence of bacterial resistance, the bacterial siderophore system has been targeted by the so-called 'Trojan horse strategy', in which antibiotics are attached to siderophore. Thus, as the bacterial cell transports the iron-loaded siderophore that is essential to its survival, it also takes up the antibiotic into the periplasm. This circumvents the outer-membrane barrier and increases the activity of antibiotics[25,26]. Recent preliminary results of a phase 2 trial with cefiderocol, a cephalosporin with a catechol vector, shows promising results in patients with urinary tract infection and pyelonephritia[27]. Natural siderophores are not in general easily chemically modified therefore synthetic catecholate derivatives, which mimic natural compounds but are tractable for attachment of antibiotics have been developed[28–30]. The design and optimisation of 'Trojan-horse' strategies targeting the enterobactin-dependent uptake system in *P. aeruginosa* would be greatly facilitated by the molecular description of PfeA-enterobactin complex, particularly since the molecular basis of enterobactin interaction with any transporter is unknown.

Here, we describe the crystal structure of $Fe^{3+}$-enterobactin bound to PfeA, which reveals the molecular details of the ferric-siderophore recognition at a binding site in the extracellular loops and distant from the binding site located on the plug domain previously described for other TBDTs. We have supported the validity of this site by a series of protein mutants guided by the complex structure, which were then analysed in vitro and in vivo. Structural data and molecular dynamics simulations suggest that this initial binding site is coupled to the periplasmic region of the transporter containing the domain interacting with the TonB-ExbB-ExbD inner-membrane complex. Molecular dynamics simulations confirm the presence of the previously suspected second site through which the siderophore passes on its route to the periplasm.

## Results

**Structural biology.** The structure of PfeA has been determined to 2.1 Å resolution with residues 15–721 of PfeA fitted into experimental density. PfeA has the same two domains (22-stranded β-barrel and N-terminal plug) common to the TBDTs (Fig. 1b). Extracellular loops of the plug domain and the β-barrel are denoted NL1 to NL3 and L1 to L11, respectively (Supplementary Fig. 1). The periplasmic turns connecting to β strands are denoted T1 to T10. PfeA has a 75% sequence homology (60% identity) with *E. coli* TBDT FepA (1FEP), which is also its closest structural relative with a root-mean-square deviation (r.m.s.d.) of 1.01 Å for 652 aligned Cα atoms. The *P. aeruginosa* TBDT, *Pa*PirA (5FP2, 72% homology and 60% identity) and the *Acinetobacter baumannii* homologue *Ab*PirA (5FR8, 64% homology and 48% identity)[31], both involved in catechol–siderophore uptake, superpose with a r.m.s.d. of 1.15 Å for 480 residues and 1.06 Å for 656 residues, respectively. These sequence and structural similarities are mirrored by the observation that all these TBDTs have been shown to transport ferric-enterobactin[32]. PfeA has loops (L3, L4, L7 and L10) that cover the extracellular face of the structure and limit the access to the plug domain compared with the other TBDTs (Supplementary Fig. 2a). A similar arrangement is seen in the FepA crystal structure, but the loops are not fully ordered in that structure. A space-filling analysis of this extracellular face reveals only a narrow tunnel with an approximate diameter of 3 Å (NE2 Gln482 to OE1 Glu327 and Asn268 ND2 to Ser479 OG). The tunnel leads to a polar void deeper inside the structure that is formed by conserved residues Gln67 of NL1, residues 100–106 of NL3, Gln267 and Asn268 of L3, residues 321–322 and Glu327 of L4, residues 477–479 and 482–483 of L7 and Tyr641. A molecule of ethylene glycol is located here forming hydrogen bonds with the side chain of Arg100 and backbone nitrogen of Trp103 (Supplementary Fig. 3a). There is a further smaller void deeper in the pore formed by Arg68, 98–102, 106–108, Asn315, 317–318, 320–321, Gly328, Tyr338, Ala340, Asp376, Ser378.

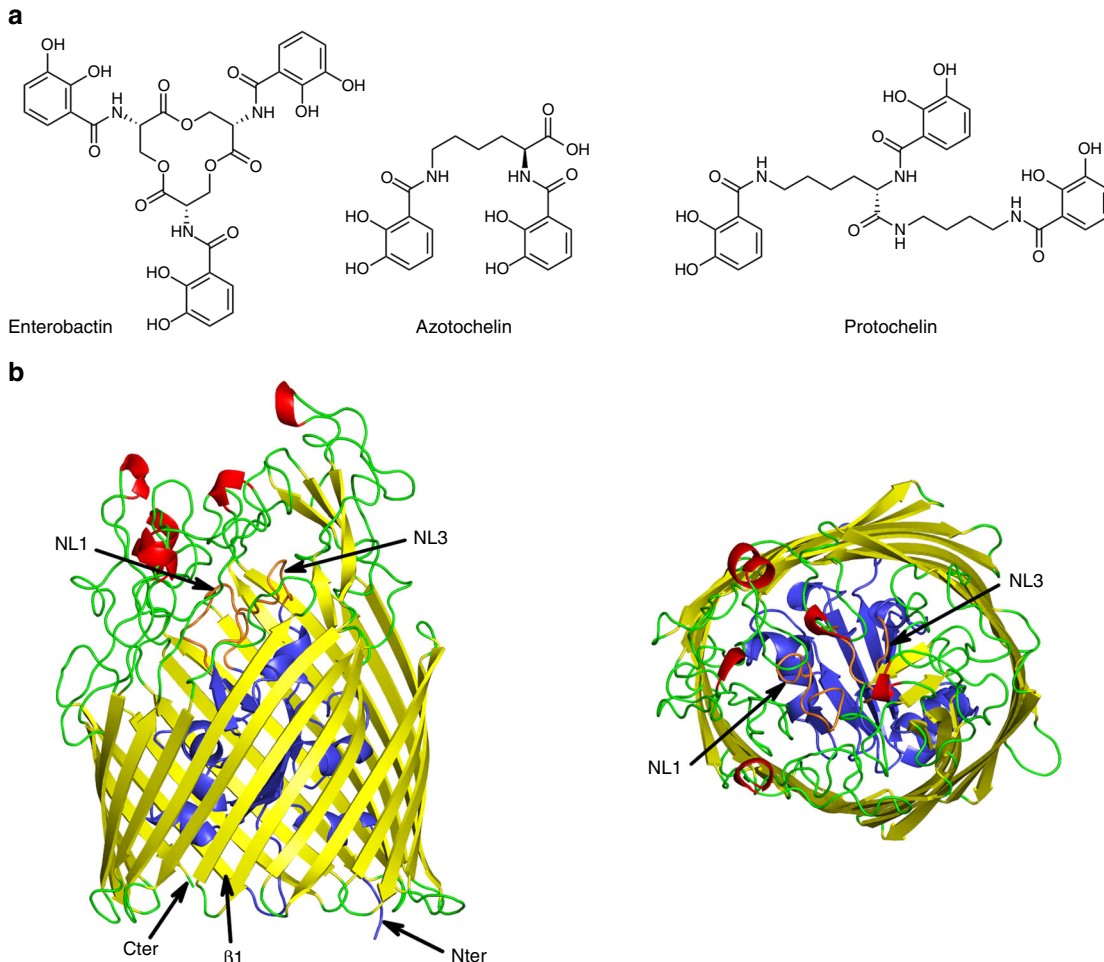

**Fig. 1** PfeA, a catecholate transporter of *Pseudomonas aeruginosa*. **a** Chemical structures of enterobactin, azotochelin and protochelin. **b** Overall structure of PfeA. The β-barrel is coloured in yellow, the loops in green, the α-helices in red, N-terminal plug domain in blue and the loops NL1 and NL3 of the plug domain in orange

Co-crystallisation efforts were unsuccessful possibly due to instability of the $Fe^{3+}$-enterobactin, which underwent colour changes and precipitated during co-crystallisation experiments. Soaking of native crystals with $Fe^{3+}$-enterobactin prior to data collection resulted in colour changes in the crystals. The data collected from such crystals revealed additional difference electron density that could unambiguously be modelled as a single $Fe^{3+}$-enterobactin molecule (Fig. 2a). The ferric-siderophore is bound to PfeA by residues from loops L2, L3, L4, L7 and L11 (Fig. 2b) on the extracellular surface, and is partly exposed to solvent. The molecule is not bound inside the cavities we identified in the apo-structure rather it is located in the entrance of the polar cavity that binds ethylene glycol. For ease of discussion, we split the enterobactin molecule into three catechol rings that we number I, II and III (Fig. 2c). The side chain of Arg480 sits between rings I and II and makes electrostatic/cation-π interactions with ring II. The side-chain of Gln482 sits between the catechol rings I and III whilst loop L4 sits between rings II and III. The catechol ring III is buried from solvent by the protein, the other two rings are partially exposed to solvent (Supplementary Fig. 2b, c). Each ring makes specific hydrogen bonds to the protein (Fig. 2c; Supplementary Fig. 4a). The oxygen atoms O6 and O3 of catecholate III hydrogen bond to Ser479 and Gln482; O5 and O2 of the catecholate II hydrogen bond to the main chain nitrogen atoms of Arg480 and Gly325, whilst O4 of catecholate I hydrogen bonds to backbone nitrogen atom of

Gln482. In addition, both Gln482 and Gln219 make hydrogen bonds with the ester of the trilactone backbone. As well as hydrogen bonds, there are extensive van der Waals contacts involving asparagine residues 268 and 270, glycines 481 and 483, valine residues 695 and 696 and Ala323, Gly324, Thr326 of loop L4 within the binding pocket. There are no π–π stacking interactions in this binding site. The more confidently identified of the two iron atoms (using anomalous diffraction) in the FepA structure was located in the vicinity of Lys483 (Arg480 in PfeA) and therefore occupies a similar position to that observed in our structure. Lys483 of FepA was also confirmed to be involved in the binding by mutation studies[33]. The second iron site in the FepA structure was identified as close to the first iron, but it was not described in detail[14]. Superposition of PfeA-$Fe^{3+}$-enterobactin structure with the previously reported FhuA-$Fe^{3+}$-ferrichrome (and related TBDT structures) show that these other transporters bind their cognate siderophores in a different location than observed here in PfeA (Supplementary Fig. 5).

The polar void described in the apo-structure is also seen in the $Fe^{3+}$-enterobactin complex structure. This void is connected to the enterobactin binding site and is formed by Gln67 of NL3, Trp103, Arg104 of NL3, Gln267, Asn268, Glu327, Leu477, Tyr641. Comparison of the $Fe^{3+}$-enterobactin complex structure with the apo-structure shows that L4 has undergone a large change (6.3 Å shift between Cα of G325) of conformation in order to interact with enterobactin molecule (Fig. 2d). This loop

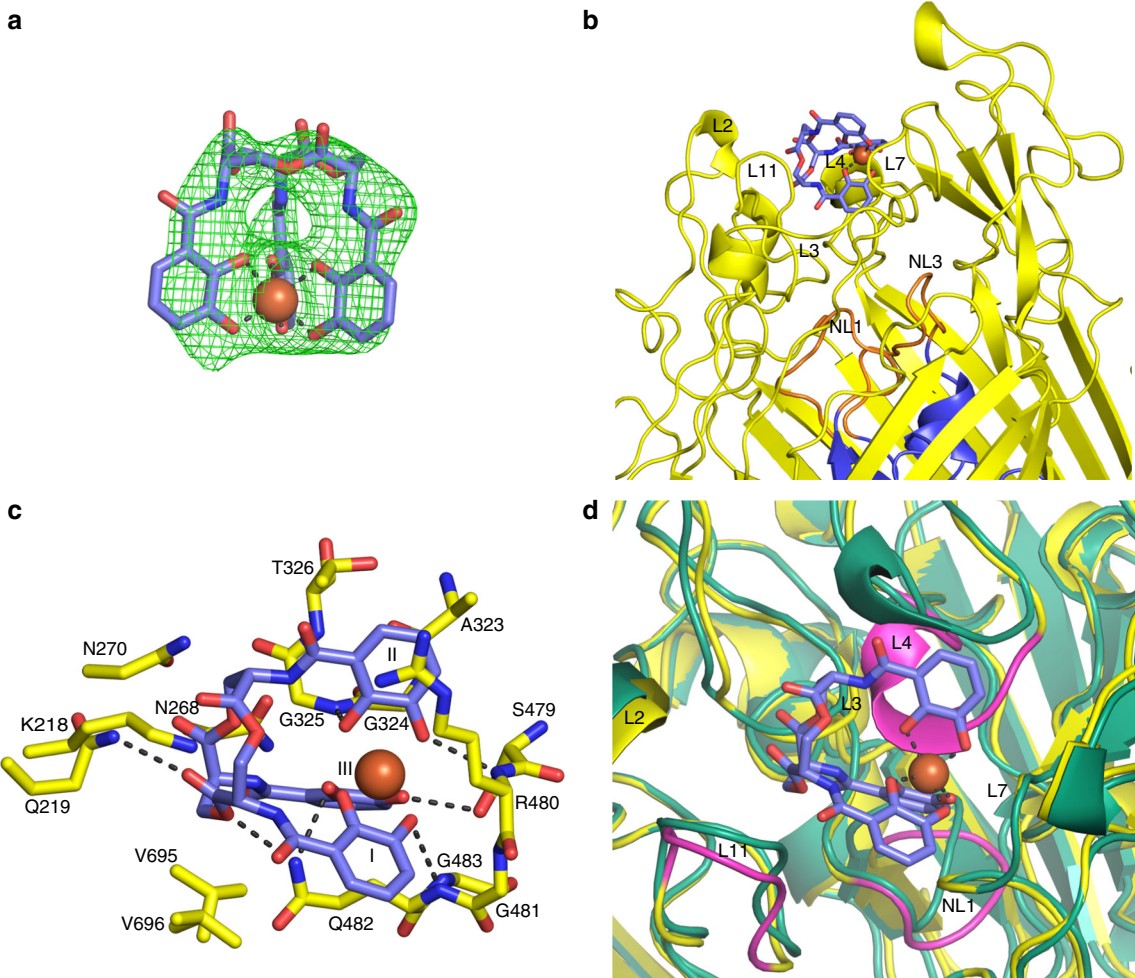

**Fig. 2** Complex structure of PfeA with $Fe^{3+}$- enterobactin. **a** $F_O$–$F_C$ electron density omit map at 3 σ around $Fe^{3+}$-enterobactin complex. Enterobactin is shown as sticks with carbon atoms coloured in blue, nitrogen in dark blue and oxygen in red. The $Fe^{3+}$ is represented as an orange sphere. **b** $Fe^{3+}$-enterobactin binds to the extracellular loops. The N-terminal plug domain is coloured in blue, the β-barrel in yellow and the NL1 and NL3 loops in orange. Secondary structure elements involved in the binding site have been labelled. **c** Binding site of the siderophore. Residues within 4.0 Å of the siderophore are displayed and hydrogen bonds are shown as black broken lines. **d** Comparison of the apo (green) and complex (yellow) structures. Loops NL1, L4, and L11 which undergo a large change of conformation have been highlighted in magenta

contains multiple glycines ($^{321}$GLAGGTEG$^{328}$) that are well known to confer structural flexibility. Three of these glycines are conserved in FepA, *Ab*PirA and *Pa*PirA (G321, G324 and G328). The change in conformation of Glu327 of L4 results in the side-chain reaching further inside the pore to make a salt bridge with Arg100 (Fig. 3a). Since the size and shape of the polar void changes upon ligand binding due to the movement of Glu327, we term this region the 'gateway'. Ring III of the enterobactin points into the 'gateway'. The gateway is connected to what we now term the internal cavity (similar to the second smaller void in the apo-structure) which is formed by Arg68, residues 99–102, 106–109, Gln267, Glu320, Leu322, 327–328, Tyr338, 378–379. In the other transporters, the cognate ligand is bound at the interface of the gateway and the internal cavity.

Concomitant with the formation of the Glu327-Arg100 salt bridge, Glu106 of NL3 (which in the apo-structure is salt-bridged to Arg100) adjusts its position. The shift of Glu106 is accompanied by a movement of 3.1 Å of NL1 of the plug domain centred on Ser63 (Fig. 3a). We also observe conformational changes in the N-terminal portion of the protein (Fig. 3b). In the apo-structure, residues 20–26 form an α-helix, and the TonB box motif ($^{13}$EQTVVATAQ$^{21}$) that engages TonB is mostly well ordered. In the $Fe^{3+}$-enterobactin complex, the peptide bond at Pro28

(β-turn type II) flips resulting in the formation of an anti-parallel β-sheet rather than the α-helix and in disorder of the TonB box. Changes in conformation, but different in detail, at the N-terminus were observed in BtuB, FecA and FhuA complexes[22,34,35], but no route for any conformational coupling with siderophore recognition was detected. We do see a correlation of movements between the binding site of the $Fe^{3+}$-enterobactin molecule in the extracellular loops, the top of plug domain and the N-terminus. This suggests a route by which siderophore binding could be coupled to TonB engagement.

**Binding of azotochelin and protochelin.** Azotochelin and protochelin are two siderophores produced by *Azotobacter vinelandii*, an environmental Gram-negative bacterium. These two secondary metabolites are, respectively, a bis-catechol siderophore and a tris-catechol siderophore[36] (Fig. 1a). It has been previously shown that in *P. aeruginosa*, PfeA is involved in the uptake of the ferric form of these siderophores, neither of which are produced by *P. aeruginosa* itself, thus *P. aeruginosa* can operate a siderophore 'piracy strategy' to acquire iron[28]. $Fe^{3+}$-enterobactin, $Fe^{3+}$-azotochelin and $Fe^{3+}$-protochelin coelute with PfeA on a size-exclusion column suggesting each of them are bound by PfeA (Supplementary Fig. 6). Crystal

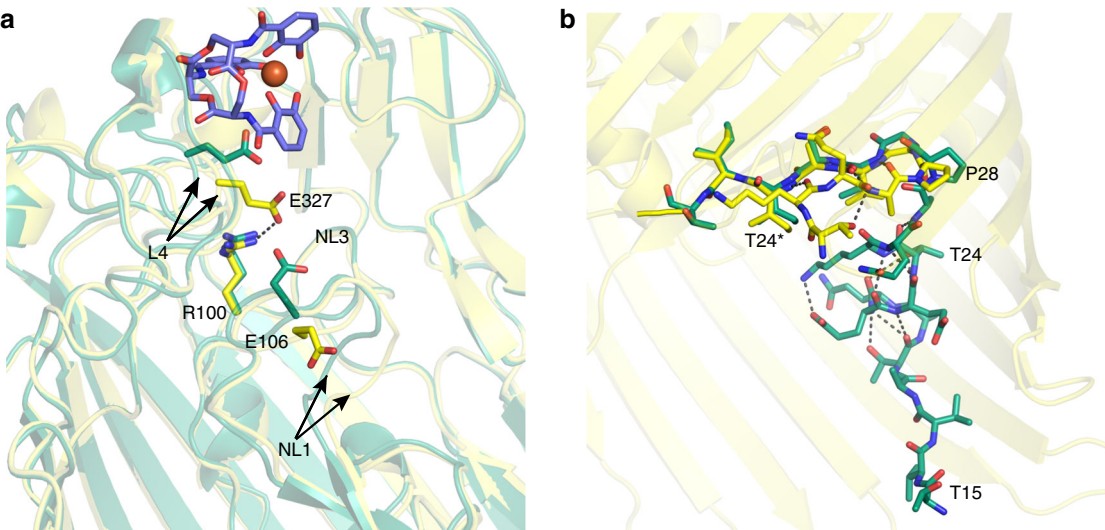

**Fig. 3** Conformation changes induced by the $Fe^{3+}$-enterobactin binding. Apo and complex structures are represented in green and yellow, respectively. **a** In the extracellular side, the movement of the loop L4 allows Glu327 to interact with Arg100 and push away Glu106 of NL3 that displaces loop NL1. **b** In the periplasmic side, the N-terminal part of PfeA rearranges following the $Fe^{3+}$-enterobactin binding. The peptide bond of the Pro28 flips resulting in the formation of β-sheet rather the α-helix observed in the apo-structure. In consequence, the TonB box become disordered. T24* refers to the threonine 24 of the complex structure

structures show that these two molecules bind in the same location as enterobactin (with less than 0.5 Å shift between the relative position of the iron atoms) and two of the catecholate rings occupy the same positions as rings II and III of enterobactin (Fig. 4; Supplementary Fig. 3c, d, Supplementary Fig. 4c, d). Hydrogen bonds between the catecholates and Ser479, Gly325 and Arg480 are conserved. In the azotochelin complex, a molecule of ethylene glycol from the crystallisation solution completes the coordination shell of the iron, and this molecule interacts with Gln482. The stacking interaction between Arg480 and catecholate II has been conserved. However, the side chain of Gln482 is unable to form a hydrogen bond with the aliphatic backbone of azotochelin, and is slightly displaced. The carboxylate group of the backbone is pointing towards the surface of the protein in the vicinity of Arg480 and Lys218. In protochelin, the presence of one supplementary carbon atom in the backbone between ring I and III perturbs the interaction with Arg480.

**Site-directed mutagenesis and biochemical characterisation**. We constructed four mutants R480A, Q482A, G324V and R480A-Q482A (double mutant) to validate the observed recognition site. Each of the mutant proteins was purified as described for the parent protein, and crystal structures of each were obtained. These structures were essentially identical to that of the parent protein, which confirmed that the mutations did not affect the folding of the protein (Supplementary Fig. 7a). Co-complexes with $Fe^{3+}$-enterobactin were obtained by soaking for the R480A and Q482A mutants (Supplementary Fig. 3e, f, Supplementary Fig. 7b, c). Apart from the mutation itself and an additional water molecule in R480A, the mutant complexes were essentially identical except that the B-factors of the $Fe^{3+}$-enterobactin molecules were higher reflecting a bigger mobility or a partial occupancy of the ligand. Neither G324V nor the double mutant were able to form a complex with $Fe^{3+}$-enterobactin (Supplementary Fig. 3g, h). Unlike the parent protein, the R480A-Q482A double mutant does not coelute with $Fe^{3+}$-enterobactin (Fig. 5a).

Titration of $Fe^{3+}$-enterobactin into PfeA by isothermal titration calorimetry (ITC) reveals an unusual biphasic behaviour (Fig. 5b; Supplementary Fig. 8, Supplementary Data 1–10) that cannot be fitted by a simple one binding site model. This result can be fitted with a cooperative two binding sites per protein model. The sites have different affinities for the ligand, one with high affinity and highly favourable enthalpy and the second with lower affinity and unfavourable enthalpy. Although the biphasic nature of the isotherm was reproducible, the numerical values varied. For K1, the binding constant was 20 to 100 nM; for K2 190 to 120 μM. The same titration was carried out with G324V and R480A-Q482A, in both these mutants no binding was detected. Mutant Q482A showed biphasic isotherms, but with much reduced heat output and smaller K1 values. Mutant R480A showed a typical low affinity-binding curve profile. The weaker binding at the first site is consistent with the higher B-factor of the $Fe^{3+}$-enterobactin molecule observed in the complex crystal structures.

**Effects of PfeA mutation on iron uptake**. To assess the effect of mutation in vivo on the transport and accumulation of iron, we used a *P. aeruginosa* strain in which the PfeA gene was deleted (ΔpfeA). This knockout strain was transformed with a plasmid containing a gene coding for the double-mutant PfeAR480A-Q482A (ΔpfeA(pMMBpfeAR480AQ482A)), and assessed with a $^{55}Fe$ uptake assay. We also measured accumulation of iron in the wild-type PAO1 strain, ΔpfeA cells transformed with native PfeA protein (ΔpfeA(pMMBpfeA)) and ΔpfeA cells with no plasmid. Cells were incubated with or without the presence of carbonyl cyanide *m*-chlorophenyl hydrazine (CCCP, an uncoupler of the proton-motive force known to inhibit all TonB-dependent transporters)[37]. In a first step, the mRNA levels of the PfeA protein and double mutant were determined to check the transcription levels of *pfeA* in all strains used for the $^{55}Fe$ uptake assay.

The mRNA levels for *pfeA* gene were 160-fold and 100-fold higher for the two strains carrying the plasmids pMMBpfeA and pMMBpfeAR480AQ482A, respectively, compared with the wild-type strain PAO1 (Fig. 6a; Supplementary Data 11). Sodium dodecyl sulfate-polyacrylamide gel electrophoresis (SDS-PAGE) of outer-membrane preparations shows that both the native and double-mutant proteins are clearly present in the outer

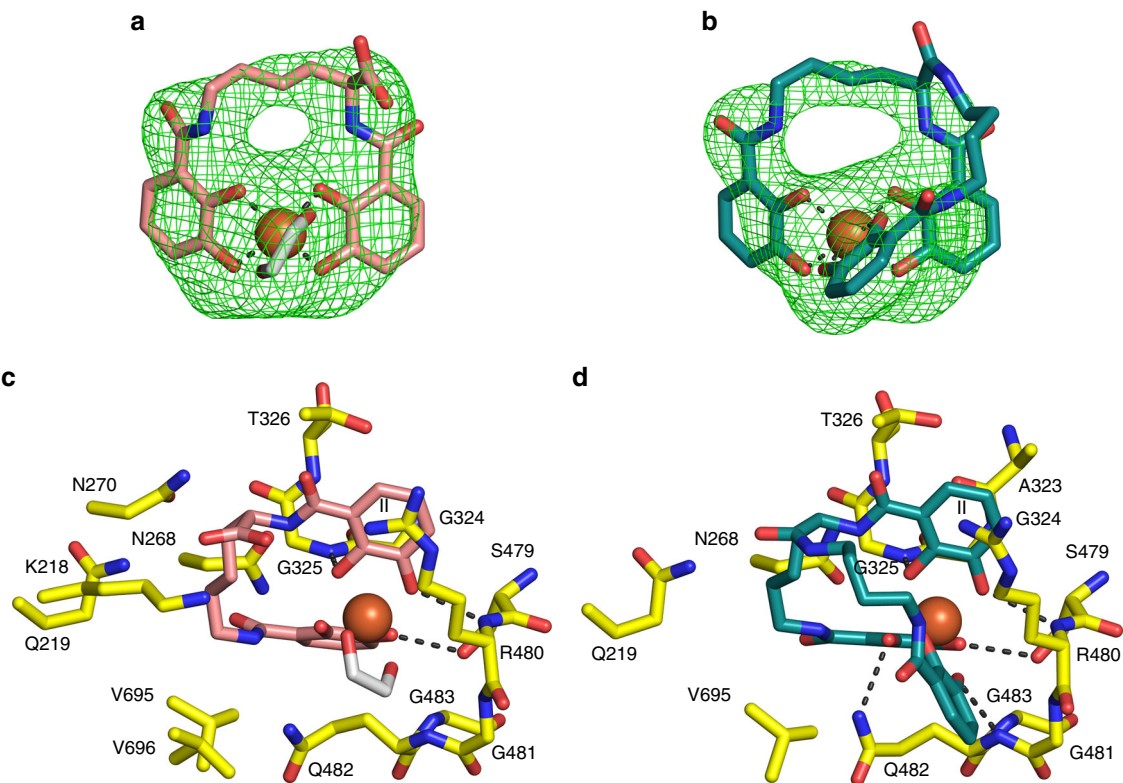

**Fig. 4** PfeA recognises the catecholate siderophores azotochelin and protochelin. $F_O$–$F_C$ electron density omit maps at 3 σ around $Fe^{3+}$-azotochelin (**a**) and $Fe^{3+}$-protochelin (**b**). Azotochelin is shown as sticks with carbon atoms coloured salmon, and protochelin carbon atoms are in green. $Fe^{3+}$ are represented as orange spheres. Binding sites of $Fe^{3+}$-azotochelin (**c**) and $Fe^{3+}$-protochelin (**d**) are the same as $Fe^{3+}$-enterobactin. Residues within 4.0 Å of the siderophores are displayed. Residues of the proteins are represented as sticks with carbon atoms coloured in yellow, nitrogen in dark blue and oxygen in red. Hydrogen bonds are shown as black broken lines. In **c**, the molecule of ethylene glycol completing the coordination shell of the iron is represented with white sticks

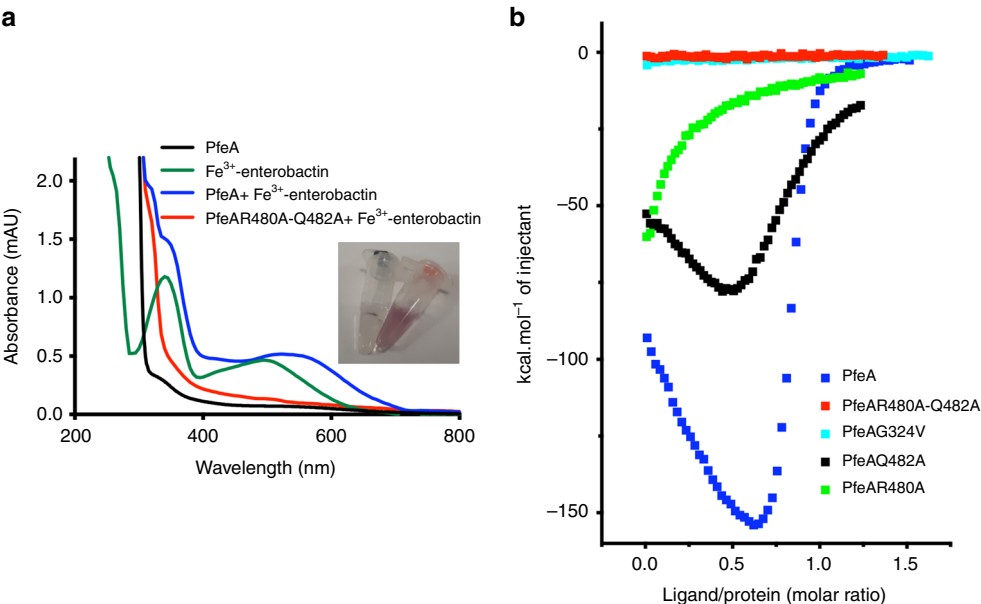

**Fig. 5** The double-mutant R480A-Q482A abolishes $Fe^{3+}$-enterobactin binding. **a** Unlike the wild-type protein (blue line), the double-mutant R480A-Q482A (red line) does not coelute with $Fe^{3+}$-enterobactin. Proteins were incubated with the $Fe^{3+}$-siderophore before being loaded on a S200 size-exclusion column. UV-visible spectra of PfeA shows a peak of absorption around 550 nm characteristic of the iron-siderophore complex. **b** Isothermal calorimetry titration of $Fe^{3+}$-enterobactin with PfeA (blue) and mutants R480A-Q482A (red), G324V (cyan), R480A (green) and Q482A (black)

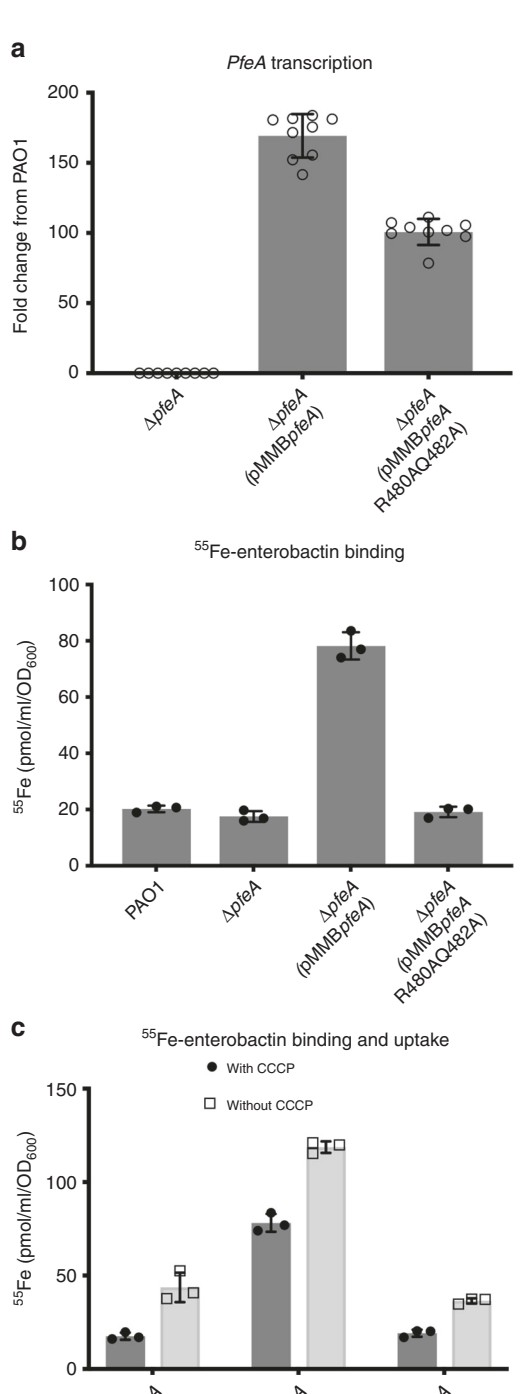

**Fig. 6** Effects of PfeA mutation in vivo. **a** Analysis of changes in the transcription of *pfeA* gene. PAO1 strain, its corresponding deletion mutant Δ*pfeA* and deletion mutants carrying the expression plasmid of PfeA wild-type (Δ*pfeA*(pMMB*pfeA*) or the R480A-Q482A mutant (Δ*pfeA*(pMMB*pfeA*R480AQ482A)) were grown in CAA medium supplemented with 10 μM enterobactin. The data were normalised relative to the reference gene *uvrD*, and are representative of three independent experiments performed in triplicate (*n* = 3). The results are given as a ratio between the values obtained for Δ*pfeA*, Δ*pfeA*(pMMB*pfeA*) and Δ*pfeA* (pMMB*pfeA*R480AQ482A) over those obtained for the PAO1 strain. **b** $^{55}$Fe-enterobactin binding at the bacterial surface. Cells were grown as in panel **a**, and were incubated for 15 min with 200 μM CCCP before initiation of transport assays by the addition of 500 nM $^{55}$Fe-enterobactin. After 30 min incubation, the radioactivity accumulated in the bacteria was counted. The results are expressed as pmol of $^{55}$Fe-enterobactin bound per ml of cells at an $OD_{600}$ of 1. The experiments have been repeated three times. **c** $^{55}$Fe-enterobactin binding and uptake. Cells were grown as in panel **a**, and were incubated with or without 200 μM CCCP before initiation of transport assays by the addition of 500 nM $^{55}$Fe-enterobactin. After 30 min incubations, the radioactivity accumulated in the bacteria was counted. The results are expressed as pmol of $^{55}$Fe-enterobactin bound and transported per ml of cells at an $OD_{600}$ of 1. The experiments have been repeated three times

outer membrane to reliably report protein binding and transport activity.

CCCP treated cells (Fig. 6b), which report only $^{55}$Fe-siderophore binding since transport is blocked, show that the knockout cells transformed with pMMB*pfeA* plasmid exhibit much higher levels of binding when compared with wild-type PAO1. This would be expected from the much higher level of PfeA protein in the outer membrane of the knockout cells (with native PfeA) than in wild-type PAO1 observed by SDS-PAGE. Knockout cells lacking a plasmid for *pfeA* show the same low level of binding as wild-type PAO1. This background level reflects the presence of other catecholate-binding proteins (e.g., PirA, CirA) present in both knockout cells and wild-type PAO1. Knockout cells transformed with double-mutant PfeA (Δ*pfeA* (pMMB*pfeA*R480AQ482A)) show the same (within error) background level of $^{55}$Fe binding. Thus, presence of double-mutant PfeA, even though expressed at a very high levels (greatly facilitating detection), does not show any detectable increase over knockout cells with no PfeA in the amount of enterobactin binding. Consequently, we conclude that the double mutant does not bind enterobactin in vivo, echoing the conclusion from the in vitro work.

Knockout cells that are not treated with CCCP report on binding and transport combined (Fig. 6c; Supplementary Fig. 10). In the knockout strain containing no plasmid, there is a small but measurable increase in the amount iron bound (compared to CCCP treated). This reflects the active transport by the other catecholate-binding proteins (e.g., PirA, CirA) and establishes the background level for the knockout system. Knockout cells expressing PfeA show a much larger increase in the amount of iron bound and a higher level of total iron than knockout cells with no plasmid. This indicates that native PfeA increases the iron level in the knockout strain because enterobactin is being actively taken up by the native plasmid encoded PfeA protein. Knockout cells transformed with the double-mutant PfeA gave essentially identical results to the knockout strain without any plasmid, that is the presence of significant levels of the double-mutant PfeA in outer membrane does not increase uptake over background. We conclude that the double mutant PfeA is not competent for the uptake of enterobactin in vivo. These data provide strong evidence that the site identified by crystallography

membrane of plasmid transformed cells (Supplementary Fig. 9). These data echo the heterologous expression of PfeA and mutants in *E. coli* used for the structural studies where the proteins are purified from the *E. coli* outer membrane at similar levels. Visual comparison of the SDS-PAGE gels (Supplementary Fig. 9) show much higher levels of both native and double-mutant proteins in the outer membrane of knockout cells (with corresponding plasmid) than the PfeA level in wild-type PAO1; this is consistent with the RT-qPCR data. We are therefore confident that the plasmid containing knockout cells, possess more than sufficient quantities of native and double-mutant PfeA proteins in their

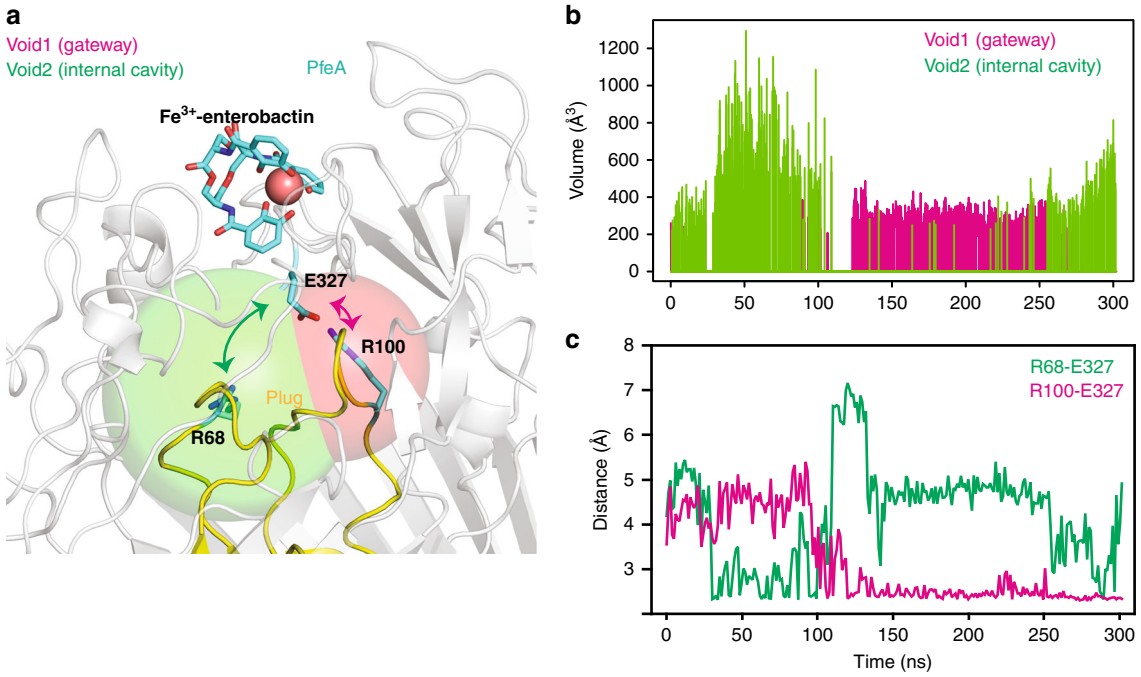

**Fig. 7** Voids identified by molecular dynamics simulations. **a** Molecular graphics illustrating gateway (magenta) and internal cavity (green) controlled by exchange of Glu327 salt bridging partners in the complex simulations. The NL1/NL3 loops of the plug region are in yellow and the rest of the protein is in white. **b** Time series of the size of the voids identified below the $Fe^{3+}$-enterobactin-binding site of PfeA. We note that they are covering almost the entire trajectory. In some parts of the trajectory, a significant expansion of the internal cavity is observed. **c** Plug-Glu327 salt bridges that control the dynamical behaviour of the internal cavity. We observe the correlation between the volume expansion and breaking and forming of the Arg68-Glu327 salt bridge

is essential for binding and transport of $Fe^{3+}$-enterobactin in vivo.

**Computational approaches**. A blind docking experiment using PfeA-$Fe^{3+}$-enterobactin structure but with the $Fe^{3+}$-enterobactin removed yielded three possible docking poses (arising from the three fold rotational symmetry) that matched the experimental complex within 0.5 Å of r.m.s.d., with an average predicted free energy of binding of $-16.7$ kcal.mol$^{-1}$ (Supplementary Fig. 11). The R480A-Q482A double mutant gave the same poses, but with a 2 kcal.mol$^{-1}$ smaller predicted free energy of binding. Analysis shows that for the double mutant, the lifetime of the $Fe^{3+}$-enterobactin complex is reduced by four orders of magnitude compared with native ($10^{-3.8}$ and $10^{0.8}$ s, respectively) indicating only a transitory interaction. Docking with the native (apo) protein structure, that is without the changes seen upon $Fe^{3+}$-enterobactin binding, yields a pose for $Fe^{3+}$-enterobactin with a much lower binding free energy (than docking with loops changed) and with a different orientation (compared to the crystal structure) (Supplementary Fig. 11).

Molecular dynamics trajectories of both the apo and ferric-siderophore complex structures were subjected to an extensive statistical analysis of internal voids[38]. In both setups, we observed two dynamic voids located between the $Fe^{3+}$-enterobactin binding site and the NL1/NL3 plug region (Fig. 7; Supplementary Fig. 12). The voids share the same residues with the internal cavity and gateway observed in the crystal structures. Although, as a result of the molecular dynamics, there are changes in positions of some of the residues that form these voids when compared with the crystal structures. Starting from the ferric-siderophore complex, the structure of the voids correlates with the hydrogen-bonding arrangement of Glu327. When Glu327 is bound to Arg100 of NL3, an arrangement resembling the gateway and internal cavity seen in the complex crystal structure is observed. An alternative arrangement, not seen in the crystal structure but in the

calculation, shows Glu327 bound to Arg68 of NL1 (Fig. 7a–c). As a result, a merged 'super' cavity is created that encompasses the gateway and the internal cavity. This super cavity has a volume compatible with the binding of $Fe^{3+}$-enterobactin ($> 500$ Å$^3$), and is closer to the plug region.

In the ferric-siderophore complex structure, we observed strong correlations between the structural changes that occur in the extracellular loops forming the binding site with the wider extracellular region and the intracellular region, notably structural changes in the periplasmic N-terminal TonB box region. Correlations are also found in the apo-structure but are less extensive (Fig. 8a, b), suggesting the presence of $Fe^{3+}$-enterobactin is important. Since the correlations are modulated by hydrogen bonds, the hydrogen bonding correlations between NL1/NL3 residues in the plug and the rest of the protein were examined in detail (Fig. 8c, d). The hydrogen bonding correlation matrix reveals a clear segregation of the hydrogen bonds in the complex, but not in the apo simulations. Thus, the binding of $Fe^{3+}$-enterobactin to PfeA creates two highly correlated groups of hydrogen bonds connecting extracellular loops to the plug domain (Fig. 8d) with Glu327 as one of the main participants.

**Discussion**

Our structural data locate a binding site for the $Fe^{3+}$-enterobactin complex distant from the plug domain, outside the barrel at the extracellular portion of PfeA formed by the loops of the barrel. This $Fe^{3+}$-enterobactin binding site involves seven hydrogen bonds between protein and ligand, as well as extensive van der Waal interactions. We note that the 120° angle between the catechol rings creates three gaps, which are filled by three conserved pieces of protein structure (Supplementary Fig. 2b). Two other siderophore complexes, azotochelin and protochelin, which are chemically different from enterobactin, utilise the same interactions. We further note that in order to form this binding

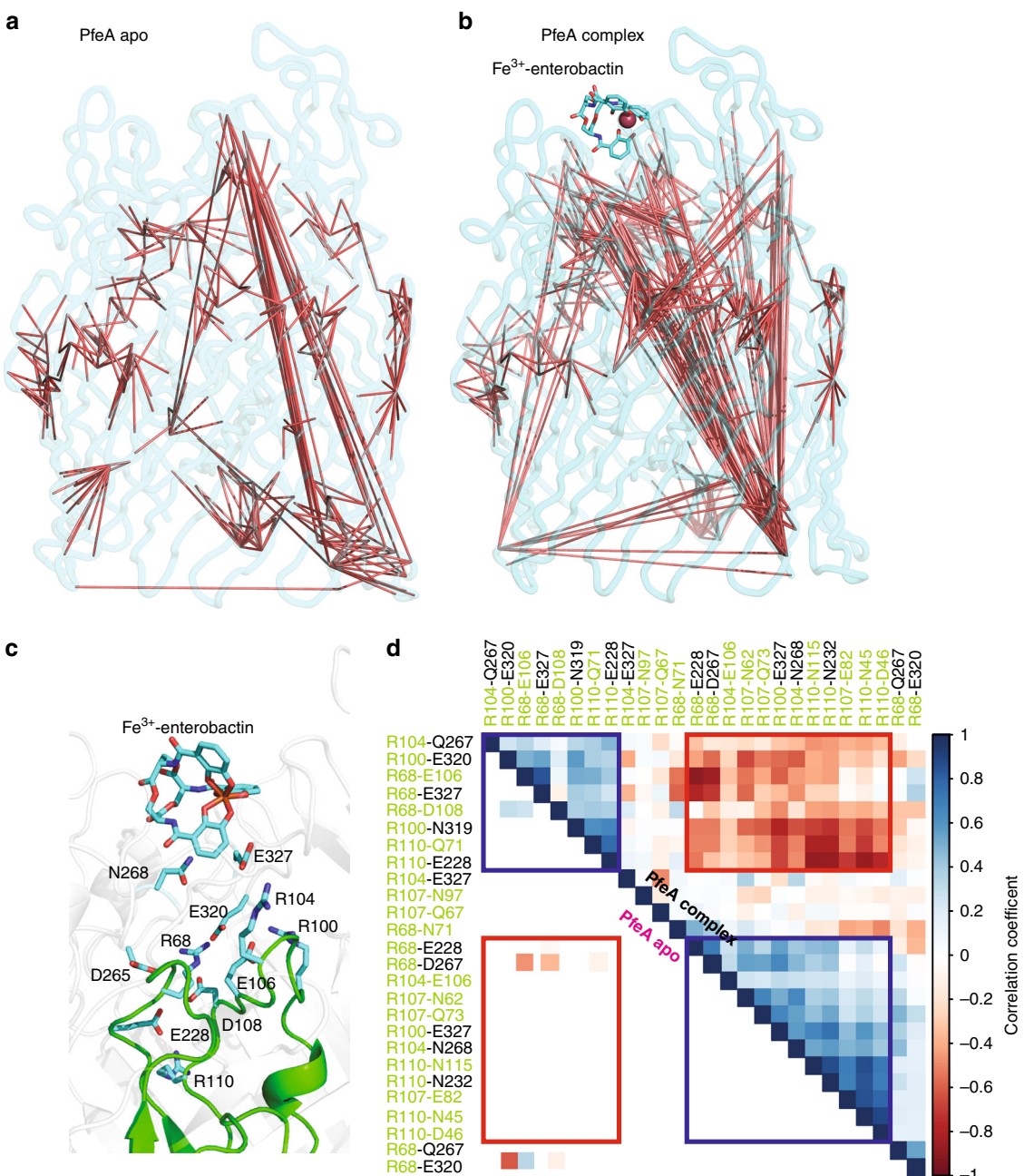

**Fig. 8** Binding of $Fe^{3+}$-enterobactin to PfeA creates two highly correlated groups of hydrogen bonds. **a**, **b** Cα correlation of complex and apo PfeA simulations, respectively. **c** Molecular graphic depicting the plug (green) NL1/NL3 amino acids that are found to be involved in hydrogen bonding with the rest of the protein (transparent white). We note that these interactions are located directly beneath the $Fe^{3+}$-enterobactin. **d** Comparison of the clustered hydrogen-bonding correlation map of the NL1/NL3 residues and the rest of the protein for apo (lower diagonal) and complex (upper diagonal). NL1/NL3 residues are marked in green. We observe formation of two dominant clusters (red and blue) of the highly correlated hydrogen bonds in complex, which are mutually anti-correlated. These correlations are absent in apo

site, the protein undergoes the same conformational changes for all three compounds and in Q482A and R480A mutants. Docking studies suggest these conformational changes are critical in providing binding energy and forming the binding site. These data are consistent with the specific recognition of a biological molecule rather than a crystal artefact.

This siderophore-binding site, relative to the protein structure, is, however, very different to other TBDT proteins where the binding site is inside the barrel atop the plug domain (Supplementary Fig. 5). We decided to probe whether the binding site in the extracellular loops was indeed genuine or an experimental

artefact of the crystal soaking strategy. ITC experiments with native protein showed nM binding affinity between PfeA and $Fe^{3+}$-enterobactin. Size exchange chromatography confirmed that the colour associated with $Fe^{3+}$-enterobactin complex migrated with the protein. These results confirm that the $Fe^{3+}$-enterobactin binds to PfeA in solution conditions similar to those in the crystal. Amino acids identified from the crystal structure as directly contributing to binding were mutated. Two single-site mutations (R480A, Q482A) of interacting residues revealed weakened (but detectable) binding in ITC experiments, behaviour predicted from the structural model and consistent with a

genuine-binding site. A R480A-Q482A double mutant and a more radical G324V mutant have the same structure as the native protein, and did not show any detectable binding to $Fe^{3+}$-enterobactin (ITC experiments, size-exclusion chromatography and crystallography). These results support the genuine nature of the binding site. Molecular dynamics calculations with the double-mutant protein indicate the complex lifetime is decreased by four orders of magnitude. When the double mutant was expressed in bacteria, it was inactive in $^{55}Fe$-enterobactin binding and uptake assays. Taken together, we regard the combination of in vivo and in vitro data as a compelling evidence that the binding location identified in the crystal structure in the extracellular loops represents a genuine and biologically important siderophore recognition site. The location of the $Fe^{3+}$-siderophore recognition site is also consistent with the iron atom position reported in the FepA crystal structure[14]. Reduced binding of siderophore was reported for a K483A FepA mutant;[33] the equivalent residue in PfeA is R480, which our structure identifies as binding to the siderophore and shows weakened binding when mutated to alanine.

If one accepts that this site distant from the plug is genuine, then it seems clear that there must be a second site closer to the plug domain. A two-site binding model has been proposed previously for FepA[13,16] with first a specific recognition by the loop followed by internalisation and uptake, but no structural data were available to support this model. The observed ITC trace for PfeA and $Fe^{3+}$-enterobactin is suggestive of a complex two-site model, one site has nM enthalpically driven binding and the other µM entropically driven binding. In the PfeA crystal structure, we identified a gateway region and an internal cavity that connect to the $Fe^{3+}$-enterobactin binding site; these regions are also found in FepA. It is in the gateway and cavity region that the other TBDT-siderophore crystal structures bind their cognate ligands, suggesting that this region in PfeA (and FepA) might indeed be a second binding site. In FepA, Trp101 and Tyr478 (Trp103 and Tyr475 in PfeA) are located in the gateway region, and have been shown to be essential for the transport[16]. Also in this region in FepA, Tyr481 (Tyr478 PfeA) has been shown to be important for both binding and transport[16]. Residues that form the cavity have been mutated in FepA, Y260A, R316A, E319A and F329A[15] (Tyr262, Arg317, Glu320, Phe330 in PfeA) and all affect transport and binding of enterobactin with Tyr260 and Arg316 having a particularly strong effect. Taking these data together points to the gateway and internal cavity forming a second siderophore-binding site.

Molecular dynamics simulations suggest that when the ligand is bound, the gateway and internal cavity are in dynamic equilibrium with a single 'super cavity', controlled by the differences in the hydrogen bonding of Glu327. Both the internal cavity and the

'super cavity' are large enough to bind $Fe^{3+}$-enterobactin. We propose a model (Fig. 9) in which the ferric-siderophore complex is recognised by the binding site observed in the crystal structure; the first binding site. Then as a result of conformation changes in the protein controlled by Glu327 the siderophore moves into the internal 'super cavity', adjacent to the plug domain; this is the second site previously proposed for FepA[15,16]. Once located in the second site, the siderophore is then ready to be translocated.

Comparing the PfeA structure with the other siderophore transporters, we suggest that the two binding site model may be a feature of the extensive loop structure of PfeA (and FepA). Only $Pa$PirA, $Ab$PirA[31] and CirA[39] are so far known to have the same large extracellular loops (particularly L7). It is worth noting that a bipartite gating process has been proposed for FecA based on the observation that extracellular loops close over the binding site to prevent the escape of ferric citrate[21], and it has been shown that FhuA loops are also important for the binding and transport of the $Fe^{3+}$-ferrichrome[40]. However, ITC and crystallography data clearly show acinetobactin receptor (BauA) (which has smaller loops) has only one binding site that is located adjacent to the plug domain (Supplementary Fig. 13)[41]. We speculate that, by soaking, we may have 'trapped' the PfeA protein in the crystal in such a way that the first binding site can form, but is unable to undergo the conformation changes needed for the siderophore complex to progress to the second site.

Our model envisages that recognition of $Fe^{3+}$-enterobactin by PfeA occurs first in the site identified in the crystal structure and since mutations of this site disrupt translocation, recognition by this first site would appear to be obligatory. If such recognition is indeed obligatory, then it will act as a filter, 'Trojan horse' antibiotic conjugates should therefore be designed to be recognised by the first site. The positioning of the three rings is clearly important in making the optimum interactions with the protein, indicating there are constraints on the linkage geometry. It is, however, worth noting that the connection of the rings tolerates the addition of at least one extra carbon, seen with protochelin and that a molecule with only two rings (azotochelin) can be efficiently bound. Further the complex structures show that the catechol rings could tolerate substituents without clashing with the protein.

Comparison of the apo and $Fe^{3+}$-enterobactin complex structures reveals a series of structural changes that connect the binding site to the plug domain and TonB box. Molecular dynamics simulations suggest that binding of $Fe^{3+}$-enterobactin is characterised by strongly correlated changes in the hydrogen-bond network of the extracellular loops and the plug domain. Such conformational coupling of the binding site, plug domain and TonB box is of course a requirement for controlled uptake. In models of translocation, the inner membrane TonB protein is

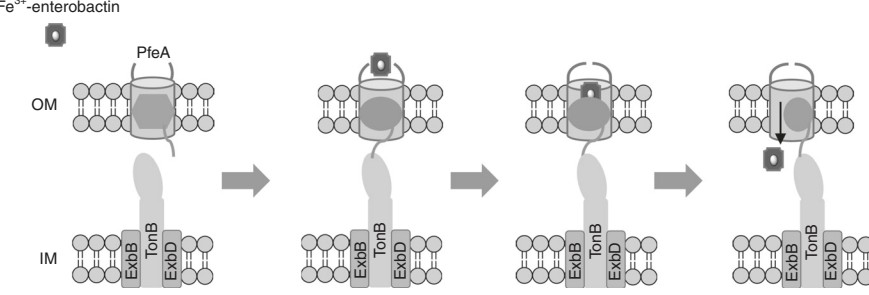

**Fig. 9** Proposed model of $Fe^{3+}$-enterobactin recognition and uptake by PfeA. $Fe^{3+}$-enterobactin is first recognised by the extracellular loops of PfeA inducing a change of conformation of the loop and the N-terminal regions of the plug domain. The siderophore can then move to the secondary binding site located in contact with the plug domain. As the result of the N-terminal conformation changes, TonB box interacts with TonB which results in a rearrangement of the plug domain that allows the translocation of $Fe^{3+}$-enterobactin

engaged by the loaded transporter, and the energy transferred by this interaction is responsible for formation of a channel through the transporter allowing the siderophore, poised in the second binding site, entry to the cell. This study reports experimental data for a structural pathway in a TBDT that communicates the siderophore binding event to the TonB-binding sequence. The results obtained here pave the way to the rational development of siderophore-antibiotic Trojan horse conjugates able to hijack more efficiently the enterobactin-dependent iron uptake system in *P. aeruginosa*.

## Methods

**Chemicals.** Azotochelin and protochelin were synthesised according to previously published protocols[28,42]. Enterobactin was furnished by Professor Carlos G. Gutierre. The protonophore CCCP was purchased from Sigma. $^{55}FeCl_3$ was obtained from Perkin Elmer Life and Analytical Sciences, in solution, at a concentration of 71.1 mM, with a specific activity of 10.18 Ci g$^{-1}$.

**Cloning.** The position of the cleavage of the signal peptide of the proteins was predicted with Signal P4.0. The coding sequence of the mature protein PfeA (PA2688) was amplified from the genomic DNA PAO1 using the primers 5′-CAAGCCATG GCCGGCCAGGGCGACGGC-3′ and 5′-AACTGGATCC TCAG AACGACGCGGTCAGGCT-3′ and KOD Hot Start DNA polymerase (Novagen). The PCR product was digested by NcoI and BamHI restriction enzymes. The gene was cloned with an N-terminal tobacco etch virus (TEV) protease cleavable His7-tag in pTAMAHISTEV vector using NcoI and BamHI restriction sites. Mutations R480A, Q482A, G324V and R480A-Q482A were constructed with a modified protocol of QuickChange method[43] using the primers
5′-CTCTACAGCGCTGGCCAGGGTTGCTACGGGCAAAG-3′ and
5′-GGCCAGCGCTGTAGAGCAGGTAGTCGGGGGTTCAGC-3′ for PfeAR480A,
5′-ACAGCCGTGGCGCGGGTTGCTACGGGCAAAGCAC-3′ and
5′-CCGCGCCACGGCTGTAGACAGGTAGTCGGGGTTC-3′ for PfeAQ482A,
5′-TCGGTGCCGACGGCCAGGCCTTCGTTGATCCGGC-3′ and
5′-GGCCGTCGGCACCGAAGGTATCTTCGACCCCAACA-3′ for G324V and
5′-TCTACAGCGCTGGCGCGGGTTGCTACGGGCAAAGC-3′ and
5′-GCGCCAGCGCTGTAGA GCAGGTAGTCGGGGGTTCAG-3′ for R480A-Q482A.

**Expression in *E. coli* and purification of PfeA.** Proteins were overexpressed in *E. coli* C43 (DE3) cells. Cells were grown at 37 °C in the Lysogeny Broth (LB) medium containing 100 µg.ml$^{-1}$ ampicillin until an optical density at 600 nm (OD$_{600}$) of 0.7, and then induced with 0.4 mM isopropyl-β-D-thiogalactopyranoside at 25 °C overnight. Details of the protein purification is described in

Supplementary Methods. In brief, isolated outer membrane pellets were solubilised with 7% (v/v) of octylpolyoxyethylene. The proteins were purified with a nickel affinity chromatography followed by cleavage of the His-tag and a second nickel affinity column. Finally, the proteins were loaded on a Superdex S200 gel filtration chromatography (10 mM Tris pH 7.5, 150 mM NaCl, 0.45% (w/v) tetraethylene glycol monooctyl ether) before being concentrated.

**Crystallisation and structure determination.** Crystals of PfeA appeared at 20 °C after few days by mixing 2 µl of protein solution (10 mg.ml$^{-1}$) with 1 µl of reservoir solution containing 11% polyethylene glycol 8000, 0.05 M ADA pH 6.5 and 0.1 M magnesium acetate. Crystals were frozen with the same solution containing 25% ethylene glycol. Enterobactin structure complexes were obtained by soaking apo crystals for few hours with mother liquor containing 5 mM Fe$^{3+}$-enterobactin before cryoprotection. For the others complexes, crystals were cross-linked by diffusion of 25% glutaraldehyde prior the soaking[44]. Co-crystallisation attempts have been performed by incubating the protein for 1 h at room temperature with the ligand before been loaded on a S200 16/60 column and concentrated for setting up the crystallisation plates. No crystals have been obtained by co-crystallisation. Crystallisation conditions of the mutants were the same as for the wild-type.

The data were collected at 100 K temperature at the beamlines IO3 (0.976 Å), IO4 (0.979 Å), IO4-1 (0.917 Å), I24 (0.968 Å) at the Diamond light source Oxfordshire or in house using a Rigaku MicromaxTM-007HF Cu anode (1.54 Å) with VariMax optics and Rigaku Saturn 944+ CCD detector. The data were processed with XIA2[45–49] autoproc[50] or HKL2000[51]. The apo-structure of PfeA was solved by molecular replacement using the coordinate of FepA (1FEP)[14] as a model with the programme PHASER[52]. Structures of the complexes and mutant proteins were solved using the apo-structure. Models were adjusted with COOT[53], and refinement was carried out using REFMAC in the CCP4 programme suite with TLS parameters[54]. Prosmart was used for the low-resolution refinements[55]. Coordinates and topologies of ligands were generated by PRODRG[56]. A large extra density corresponding probably to a lipid is present on the surface of the protein. Due to high mobility of the molecule, only two acyl chains have been modelled in the enterobactin structure. Final refinement statistics are given in the Table 1. Atomic coordinates and structure factors have been deposited in the Protein Data Bank (5M9B, 6Q5E, 5MZS, 5NC4, 5NR2, 5OUT, 6I2J and 6R1F). The quality of all structures was checked with MOLPROBITY[57]. There are no Ramachandran outliers in either of the structures and at least 96% of the residues are in the most favoured region. Figures were drawn using PYMOL[58].

**Isothermal microcalorimetry titration.** Affinities of PfeA wild-type and mutants for Fe$^{3+}$-enterobactin were measured by isothermal titration calorimetry using a VP-ITC instrument (GE Healthcare) at 25 °C. Titrations were performed using 57 × 5 µl injections of Fe$^{3+}$-enterobactin (70–55 µM) into 10 µM protein. The heats of dilution measured from injection of the ligands into the buffer were subtracted, and titration curves were fitted using AFFINImeter software with a two-site binding model. Concentrations of Fe$^{3+}$-enterobactin solutions were verified with

## Table 1 Crystallographic data and refinement statistics

| | Apo (5M9B) | Enterobactin complex (6Q5E) | Azotochelin complex (5NR2) | Protochelin complex (5NC4) | R480A-Q482A (5MZS) | G324V (5OUT) | R480A (6R1F) | Q482A (6I2J) |
|---|---|---|---|---|---|---|---|---|
| *Data collection* | | | | | | | | |
| Space group | P2$_1$2$_1$2 | P2$_1$2$_1$2 | P2$_1$2$_1$2 | P2$_1$2$_1$2 | P2$_1$2$_1$2 | P2$_1$2$_1$2 | P2$_1$2$_1$2 | P2$_1$2$_1$2 |
| Cell dimensions | | | | | | | | |
| *a, b, c* (Å) | 87.9, 158.2, 77.8 | 85.6, 159.3, 78.6 | 86.6, 158.1, 79.0 | 86.7, 157.4, 78.1 | 90.2, 56.9, 76.6 | 87.9, 158.1, 77.9 | 86.50, 157.26, 78.00 | 87.0, 158.3, 78.5 |
| α, β, γ (°) | 90, 90, 90 | 90, 90, 90 | 90, 90, 90 | 90, 90, 90 | 90, 90, 90 | 90, 90, 90 | 90, 90, 90 | 90, 90, 90 |
| Resolution (Å) | 76.83-2.12 (2.18-2.12)* | 79.63-2.70 (2.85-2.70)* | 86.64-2.78 (2.85-2.78)* | 78.68-2.8 (2.81-2.8)* | 76.58-2.67 (2.72-2.67)* | 39.08-2.90 (2.95-2.90) | 43.51-3.11 (3.16-3.11) | 79.14-2.95 (3.00-2.95) |
| $R_{sym}$ or $R_{merge}$ | 0.052 (1.190) | 0.057 (0.603) | 0.048 (1.057) | 0.054 (0.695) | 0.105 (1.197) | 0.106 (1.377) | 0.087 (1.98) | 0.071 (0.627) |
| $I / σI$ | 20.3 (1.7) | 22.0 (2.0) | 20.8 (1.8) | 19.5 (2.3) | 15.5 (1.4) | 12.7 (1.4) | 14.2 (1.1) | 21.7 (1.6) |
| Completeness (%) | 99.6 (99.8) | 99.9 (100.0) | 99.9 (99.8) | 100 (97.6) | 99.9 (99.9) | 80.4 (81.9) | 100 (100) | 93.3 (92.3) |
| Redundancy | 7.5 (7.4) | 6.5 (6.6) | 7.3 (7.3) | 7.2 (7.7) | 6.9 (6.1) | 3.5 (3.5) | 7.5 (8.0) | 6.3 (4.5) |
| CC half | 0.999 (0.698) | – | 0.999 (0.677) | 0.999 (0.867) | 0.998 (0.662) | 0.998 (0.669) | – | – |
| *Refinement* | | | | | | | | |
| Resolution (Å) | 76.83-2.12 | 79.63-2.70 | 86.64-2.78 | 78.68-2.80 | 76.58 (2.67) | 39.08 (2.90) | 43.51 (3.11) | 79.14 (2.95) |
| No.of reflections | 59043 | 28689 | 26627 | 25602 | 30008 | 18869 | 18734 | 18979 |
| $R_{work}/R_{free}$ | 0.200/0.231 | 0.21/0.249 | 0.225/0.260 | 0.218/0.263 | 0.220 (0.266) | 0.255 (0.296) | 0.214 (0.256) | 0.302 (0.316) |
| No. of atoms | | | | | | | | |
| Protein | 5435 | 5370 | 5363 | 5363 | 5418 | 5438 | 5364 | 5366 |
| Ligand/ion | 44 | 83 | 35 | 46 | – | – | 49 | 49 |
| Water | 256 | 33 | – | – | 7 | 7 | 1 | 1 |
| *B*-factors | | | | | | | | |
| Protein | 64.80 | 80.99 | 110.11 | 111.42 | 86.5 | 71.50 | 138.80 | 100.70 |
| Ligand/ion | 62.10 | 83.47 | 93.09 | 110.01 | – | – | 139.87 | 102.00 |
| Water | 51.00 | 57.45 | – | – | 67.40 | 42.00 | 77.62 | 79.41 |
| *r.m.s. deviations* | | | | | | | | |
| Bond lengths (Å) | 0.011 | 0.007 | 0.012 | 0.012 | 0.012 | 0.007 | 0.011 | 0.011 |
| Bond angles (°) | 1.447 | 1.416 | 1.379 | 1.406 | 1.473 | 1.255 | 1.32 | 1.44 |

UV/vis spectrophotometry using a published molar extinction coefficient of $\varepsilon_{495} = 5600\ M^{-1}\ cm^{-1}$. Determination of thermodynamic parameters is not fully accurate because errors in $Fe^{3+}$ complex, active PfeA concentrations and unusual profile giving estimates of K1 = 23–100 nM and K2 = 185–128 μM for the wild-type. Reverse titration has been also performed (PfeA into $Fe^{3+}$-enterobactin), but data suffer from high heats of dilution in the beginning of the blank titration of PfeA into the buffer giving estimates of K1 = 95 nM and K2 = 4.5 μM for the wild-type.

**$^{55}$Fe-enterobactin binding and uptake assays**. The complex was prepared at $^{55}$Fe concentrations of 50 μM, with a siderophore:iron (mol:mol) ratio of 20:1. *P. aeruginosa* strains were first grown overnight at 30 °C in LB broth, and were then washed, resuspended and cultured overnight at 30 °C in iron-deficient CAA medium (casamino acid medium, composition: 5 g.l$^{-1}$ low-iron CAA (Difco), 1.46 g.l$^{-1}$ K$_2$HPO$_4$ 3H$_2$O, 0.25 g.l$^{-1}$ MgSO$_4$ 7H$_2$O) and in the presence of 10 μM enterobactin to induce the expression of PfeA. CAA is a highly iron-restricted medium with iron concentrations of 20 nM[59]. Afterwards bacteria were washed with 50 mM Tris-HCl pH 8.0, and diluted to an OD$_{600}$ of 1. Cells solutions were divided in two, one batch being treated with 200 μM CCCP. This compound collapses the proton-motive force across the bacterial cell membranes, inhibiting TonB-dependent iron uptake[37]. Binding and transport assays were initiated by adding 500 nM $^{55}$Fe-enterobactin to the bacterial solutions treated or not with CCCP. After 30 min of incubations, cells were centrifuged, and the radioactivity associated to the bacteria was monitored in the cell pellets[60]. For the cells treated with CCCP, the radioactivity in the cell pellets corresponds to the binding of $^{55}$Fe-enterobactin to *P. aeruginosa* cell surface, since no uptake occurs. In the absence of CCCP, the radioactivity associated to the cells corresponds to $^{55}$Fe-enterobactin binding to the cell surface and the uptake into the bacteria.

**Quantitative real-time PCR**. Specific *pfeA* gene transcription was measured by reverse transcription-quantitative PCR (RT-qPCR) from 16 h cultures in CAA medium as described in Supplementary Methods.

**Outer-membrane preparations**. Bacteria were grown in the CAA medium in the presence of 10 μM enterobactin as described above for the $^{55}$Fe uptake assays. Bacterial cultures were centrifuged, cell pellets were washed twice with 50 mM Tris-HCl pH 8.0 and then resuspended in 1.5 ml of buffer A (200 mM Tris-HCl pH 8.0, 20% (w/v) sucrose). Lysozyme (Euromedex) was added to the suspension to a final concentration of 200 μg.ml$^{-1}$, and the mixture incubated at room temperature for 30 min. Spheroplasts were pelleted by centrifugation (10 min at 8,500 × g), cold water (1 ml) was added and the resulting suspension was incubated at 1 h at 37 °C with benzonase (1 μl of benzonase ≥ 250 units/μl from Sigma). Membranes were isolated by ultracentrifugation (40 min at 100,000 × g). Membrane pellets were washed twice with ultra-pure water and resuspended in 200 mM Tris-HCl pH 8.0, 1% (w/v) *N*-lauroylsarcosine buffer and incubated during 1 h at room temperature to solubilise specifically all inner-membrane proteins. Afterwards, outer membranes were pelleted by ultracentrifugation (40 min at 100,000×g).

**Computational methods**. The complete system under investigation, the PfeA receptor, was inserted in a phospholipid membrane and fully solvated by employing the CHARRM-GUI web server[61]. In particular, the lipid bilayer is composed of 233 POPC (1-palmitoyl-2-oleoyl-sn-glycero-3-phosphocholine) molecules with *xy* planar dimensions of 100Å × 100Å. The system was then immersed in an explicit water solution adding KCl ions to a concentration of 0.15 M. Molecular dynamics (MD) trajectories were produced by using the ACEMD software[62]. We chose the AMBER14 and the LIPID14 force fields[63,64], respectively, for the protein and the lipids, whereas we used the TIP3P model[65] for waters. The force-field parameters for $Fe^{3+}$-enterobactin complex were obtained using the Metal Center Parameter Builder MCPB[66]. After the initial heating, we equilibrated our system in the NPT ensemble gradually releasing constraints that had been initially applied to the protein Cα/Cβ atoms and phosphorus atoms of the lipid head groups. After equilibration, we performed 300-ns-long molecular dynamics simulations under the NVT ensemble, using as fixed volume the average volume of the previous equilibration stage. Pressure and temperature were regulated at 1 Atm and 310 K using the isotropic Berendsen barostat and the Langevin thermostat, respectively. Electrostatic interactions were evaluated using the particle mesh Ewald scheme, with a cut-off of 9.0 Å for the short-range evaluation in direct space and with frequency set to 2. The switch function is applied to non-bonded interactions with the switch-distance set to 7.5 Å and cut-off 9.0 Å. In order to accelerate the simulations, mass of hydrogen atoms were scaled to 4 uma, which allowed a time step of 4 fs.

Cavity analysis was conducted by using the VOIDOO software[67,68]. We used a probe radius of 1.5 Å (size of a water molecule) rolling over a grid with maximum bin size set to 1.0 Å. After the initial cavity identification, we refined the search by reducing the grid size in an iterative manner until a convergence of the detected cavity is not reached. We adopted a convergence criterion of 0.1% between successive volume evaluations. The entire procedure was then repeated for all conformers, which were saved at the frequency of 90 ps (~ 10,000 conformers in total or 900 ns on total time). After collecting cavity data for all of our simulations, we performed a cluster analysis imposing an r.m.s.d. of 3.0 Å.

The Cα correlation was calculated by using the bio3d package within R[69] and a statistical mechanical approach independent of relative atomic motions known as LMI (linear mutual information)[70]. The main advantage of the LMI over the standard methods for detecting correlation in protein motions is the invariance to a relative orientation of atomic fluctuations. Due to this property, LMI captures a more detailed picture of how protein motions are correlated. Moreover, LMI omits unwanted non-linear correlations and makes a perfect candidate for investigation of the protein dynamics. To test convergence, we performed a block analysis where the total trajectory is divided into 50-ns windows, and a correlation matrix was calculated for each window.

The hydrogen-bonding correlation was calculated using a specifically designed protocol. In short, first we calculate possible hydrogen-bonding interactions for every 100 ps. After a list of all possible hydrogen bonds at a given frequency is collected, we record a minimum distance profiles for each hydrogen-bonding pair. These profiles are then used to calculate a correlation between the specific hydrogen bonds. All figures and plots were produced using the VMD[71] and R[72] softwares.

**Docking**. All molecular docking calculations have been performed using AutoDock Vina[73]. We adopted a docking volume represented by a rectangular box 60 × 60 × 60Å in size, centred at (0.0, 0.0, 20.0 Å), which completely encloses the binding domain of the protein. The exhaustiveness of the search was set to 4096 (default 8). The coordinates of enterobactin have been kept frozen at the X-ray configuration, while for PfeA we used the X-ray structures as well as different snapshots extracted from MD simulations.

**Binding lifetime**. Metadynamics simulations were performed leaning on trajectories produced during NVT runs for the PfeA/$Fe^{3+}$-enterobactin system. We used ACEMD with the PLUMED2[74] plugin starting from the last snapshot of the MD trajectory. For the first collective variable, we choose the z-component of the centre of mass distance between the PfeA and $Fe^{3+}$-enterobactin. As for the second collective variable, we choose the number of contacts between the PfeA and $Fe^{3+}$-enterobactin. In order to achieve differentiability, the number of contacts is calculated employing the switching function centred on 3.0 Å that considers only the neighbouring atoms within the 5 Å cut-off. We ran 24 simulations in total, 12 for the wild-type and 12 for the double mutant. We stopped the calculations when $Fe^{3+}$-enterobactin detached from the proposed X-ray binding site. After the exit, the $Fe^{3+}$-enterobactin's lifetimes obtained with the metadynamics were translated to real residence times using the method implemented by Tiwary and Parrinello[75]. These residence times were then fitted to the Pdf, and the Kolmogorov–Smirnov test[76] was performed in order to check the accuracy.

**Reporting summary**. Further information on research design is available in the Nature Research Reporting Summary linked to this article.

## Data availability

Structures and data have been deposited with the RCSB with the following codes: 5M9B, 6Q5E, 5NR2, 5NC4, 5MZS, 5OUT, 6I2J and 6R1F. Raw ITC traces and qRT PCR raw data are available as Supplementary Data. All genes used in the structural study have been deposited with ADDGENE (pTAMAHISTEV_PfeA [https://www.addgene.org/128943/], pTAMAHISTEV_PfeAR480A [https://www.addgene.org/128944/], pTAMAHISTEV_PfeAQ482A [https://www.addgene.org/128945/], pTAMAHISTEV_PfeAR480A-Q482A [https://www.addgene.org/128946/], pTAMAHISTEV_PfeAG324V [https://www.addgene.org/128947/]). The source data underlying Figs. 5a, 5b, 6a-c, Supplementary Figs. 6a–c, 9a–c and 10 are provided as a Source Data file.

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

## Acknowledgements

The research leading to these results was conducted as part of the Translocation consortium (www.translocation.eu) and benefitted from support from ND4BB ENABLE Consortium has received support from the Innovative Medicines Initiatives Joint Undertaking under Grant Agreement nos. 115525 and 115583, resources which are composed of financial contribution from the European Union's seventh framework programme (FP7/ 2007–2013) and EFPIA companies in kind contribution. This is work is supported by a Wellcome Trust Investigator (100209/Z/12/Z) award. M.C. and S.M. thank the additional financial support of MIUR with the PRIN project 2015795S5W_005. I.S., G.L.A.M., V.G. and E.B. thank also *Vaincre la Mucoviscidose* and *Association Gregory Lemarchal*, French associations against cystic fibrosis for additional financial support. J.H.N. and L.M. thank the Membrane Protein Laboratory at Diamond for beam time access.

## Author contributions

L.M. performed all the crystallography, biophysics, analysed the data and wrote the paper. G.M. performed docking calculations and force field of enterobactin-iron complex. S.M. performed molecular dynamics simulations, analysed the data and wrote the paper. M.C. directed project, analysed the data and wrote the paper. M.G.P.P. secured funding and directed project. R.P.M. performed mutant proteins purification and crystallisation. J.H.N. directed project, analysed the data and the wrote paper. V.G. performed $^{55}$Fe uptake assay and construct some of the strains. E.B. performed azotochelin and protochelin synthesis and purification. G.L.A.M. designed the syntheses and analysed the chemical data. I.J.S. directed project, analysed the data and wrote the paper.

## Additional information

**Competing interests:** The authors declare no competing interests.

