## [Peer Review File · Nature Communications]

Reviewers' comments:

Reviewer #1 (Remarks to the Author):

This paper presents a detailed X-ray, molecular dynamics and ITC study of complexes of siderophores with WT and site-point mutations of PfeA, a membrane transporter from the pathogenic bacterium *P. aeruginosa*.

Eight X-ray structures of ligand complexes and site-point mutants have been solved and refined as part of this work. There are no major surprises in terms of new structural features as orthologous proteins have been determined previously. The new structures do however provide useful information on the detailed binding modes of the ligands and furthermore show that the mutant structures adopt properly folded conformations. All structures (apart from 5m9b) have good refinement statistics. (5m9b is a low resolution structure with an R_{free} of 35% and all four 'water molecules' have unrealistically low (B=9.7) temperature factors; all of which suggests that this structure has some issues)

The ITC data coupled with the siderophore transport properties of the mutants provide convincing evidence that the ligand binding pocket identified by the crystal soaking experiments is also biologically relevant.

Molecular dynamics simulations were carried out for 300ns which is an appropriate time for such a study. The MD results provide support for the proposed model of how the siderophore is transported through the middle of the receptor as summarised in the final figure of the paper. This paper provides a well-argued and quite detailed model of how the ligands are transported through the protein. Such a model could be useful in the design of new siderophore-conjugate therapeutics that would be able to enter gram negative bacteria. Given the current interest in developing new approaches to tackling AMR, the paper should be of broad general interest.

Some specific points in the text are given below:

Line 83 Grammar/ meaning of this sentence is unclear

Once the ferric-siderophore is bound to the receptor, depending on the siderophore and on the bacteria considered, iron will be delivered either into the periplasm or into cytoplasm.

Line 96

The coordination of Fe³⁺ by the three catecholates results in octahedral arrangement with a net charge of -3

The sentence implies that the complex has a net charge of -3

Line 113 Grammar

it simultaneously takes up the antibiotic into the periplasms;

Line 115

results in patient with urinary tract infection and pyelonephritia²⁷.

In a patient or in patients ?

Line 139

with *E. coli* TBDT FepA which is also its closest structural relative with an r.m.s.d of 1.01 Å for 652 aligned Ca atoms. The *Pseudomonas* TBDT, PaPirA (72% homology and 60% identity) and the *Acinetobacter baumannii* homologue AbPirA (64% homology and 48% identity)

It would be useful to mention relevant pdb codes in this sentence

Line 216 Grammar

Like for Fe³⁺-enterobactin, Fe³⁺-BCS and Fe³⁺-TCS coelute

Line 236

Apart from the mutation itself and an additional water molecule in R480A, the mutant complexes were essentially identical except that the density for the Fe³⁺-enterobactin molecule was weaker. Could this weakness be quantified and discussed in the text by using B-factors, occupancies, anomalous signals of the iron ions ?

L238

the parent protein, the R480-Q482A double
Should be 'the parent protein, the R480A-Q482A double '

Line 258

Protein has been incubated with the Fe³⁺-siderophore before been loaded on a

Line 282

very strong correlations between the extracellular loops that form the binding site
More discussion is need here to describe what sort of correlations and what they mean in terms of the proposed mechanism

Reviewer #2 (Remarks to the Author):

This study identifies an apparent binding motif for the siderophore enterochelin on external surface of the *Pseudomonas* FepA homologue PfeA. The solved crystal structure in the absence of enterochelin ligand differs from those of other crystallized TBDT proteins in that the internal portions of the beta barrel are obstructed by the folding of external loops, leaving a small hole. Crystals soaked with enterochelin take up the siderophore, with those structures then showing enterochelin bound to residues comprising edge features of this small hole. Mutations in these residues partially support the contention that this is a biologically relevant binding site.

The central issue is whether or not this is real. The authors raise this question (Lines 297-298) "whether this new binding site in the extracellular loops was indeed genuine or an experimental artefact". This is followed by (Lines 342-343) with "We speculate that, by soaking, we may have 'trapped' the PfeA protein in the crystal in such a way that the first binding site is formed..." So yes, it looks like it is trapped in a specific conformation - the authors note that the loops in the *E. coli* FepA structure are more disordered what they see and that the π - π stacking found to be important in vivo is not evident (lines 174-176). I think the authors recognize this, but the way everything is described suggests that this is the normal conformation, ligand binds, and then is moved into the internal chamber where it can then associate with the cork domain. What is not mentioned here is that we have known for a long time that TBDT ligands do bind to regions in the outer loops - work in the Braun lab in the 1990s (J. Bact. 174: 3479 & 177: 694) identified loop regions in FhuA required for phage and colicin M docking that could be competed for with the siderophore ferrichrome and the siderophore-conjugated antibiotic albomycin. The idea of surface-exposed binding domains that first capture free ligands to enhance interaction with a second domain that mediates an energy-dependent event is well established and widespread in molecular systems. This study identifies some residues that are important, and can "catch" diffusing enterochelin, even when locked into a rigid crystal structure. That does not mean this is a conformation that exists in vivo. Mutations could disrupt that binding, and residue G342 was essential (as a glycine probably more about structure than as a site itself) - R480 and Q482 are better candidates for interaction - and as the authors note (line 316) the residue corresponding to R480 in FepA (misidentified as "K83") has been shown to be important for enterochelin binding. In summary, this is a crystal structure with surface exposed residues that have a role in binding

enterochelin. That does not mean this is a structure that exists in vivo - there are many other potential structures in which those residues would be accessible.

As a related point, the studies with azotochelin and protochelin (Fig 4) do not address the question of whether or not the solved structure is relevant - they are simply in silico predictions consistent with the the above in vitro observation.

The second major point involves the iron transport assays (Fig. 6). All of the appropriate controls are present, and so the focus is on the comparison of the plasmid-borne wild-type (pMMBpfeA) and the double mutant (pMMBpfeAR480AQ482A). The unresolved question here is "are the slopes different?" Baseline for the wild type is ~80 and it climbs to ~120 (why does it start so much higher than all the others? - was it more highly expressed so you have more loading prior to transport - did you check expression levels?); the double mutant baseline is around 25 but drops to 20 and then is at 50 and ends at 40. So the transport is a little less than a doubling of ligand over the 30 minutes. Same is true for the wild type. Statistics would be useful here (what are the slopes and what are their r values? "...experiments have been repeated three times and equivalent kinetics have been observed" (line 567) is not helpful.

Other aspects to address are:

- 1.) There is a general lack of qualification of statements. For example (line 334, line 362) "Molecular dynamics shows..." No it doesn't - models suggest - they do not show.
- 2.) There is a wide use of abbreviations and acronyms without definition.
- 3.) There are a number of typos and grammatical issues that can be readily cleaned up - but be careful with aspects that aren't evident, like the error on line 316 with the FepA residue (see above).

In closing, this is a good study, but over-interpreted...

R.A. Larsen

We thank the reviewers and editors for their helpful comments that clearly identify areas for improvement in the original manuscript. We believe this revised manuscript is significantly improved. Our details responses are set out in red. A tracked changes document has been uploaded. Review comments are shown in blue.

Reviewer #1 (Remarks to the Author):

This paper presents a detailed X-ray, molecular dynamics and ITC study of complexes of siderophores with WT and site-point mutations of PfeA, a membrane transporter from the pathogenic bacterium *P. aeruginosa*.

Eight X-ray structures of ligand complexes and site-point mutants have been solved and refined as part of this work. There are no major surprises in terms of new structural features as orthologous proteins have been determined previously. The new structures do however provide useful information on the detailed binding modes of the ligands and furthermore show that the mutant structures adopt properly folded conformations. All structures (apart from 5m9b) have good refinement statistics. (5mb9 is a low resolution structure with an R_{free} of 35% and all four 'water molecules' have unrealistically low (B=9.7) temperature factors; all of which suggests that this structure has some issues).

We think the reviewer refers to 6i2l instead of 5m9b. We agree the quality of the data was poor. We have obtained new data and this entry has been replaced by a new structure (6R1F) with a better resolution (3.11) and better statistics (R_{free}=0.246). Text and table have been updated.

The ITC data coupled with the siderophore transport properties of the mutants provide convincing evidence that the ligand binding pocket identified by the crystal soaking experiments is also biologically relevant.

Molecular dynamics simulations were carried out for 300ns which is an appropriate time for such a study. The MD results provide support for the proposed model of how the siderophore is transported through the middle of the receptor as summarised in the final figure of the paper.

This paper provides a well-argued and quite detailed model of how the ligands are transported through the protein. Such a model could be useful in the design of new siderophore-conjugate therapeutics that would be able to enter gram negative bacteria. Given the current interest in developing new approaches to tackling AMR, the paper should be of broad general interest.

Some specific points in the text are given below:

Line 83 Grammar/ meaning of this sentence is unclear

Once the ferric-siderophore is bound to the receptor, depending on the siderophore and on the bacteria considered, iron will be delivered either into the periplasm or into cytoplasm.

The referee is right, we apologise, the sentence has been modified:

“Once the ferric-siderophore has been transported across the outer-membrane, iron will be released either into the periplasm or into cytoplasm, depending on the siderophore and on the bacteria considered”

The coordination of Fe³⁺ by the three catecholates results in octahedral arrangement with a net charge of -3. The sentence implies that the complex has a net charge of -3.

The literature says that both hydroxy groups of the three catechols are deprotonated giving a charge of -6 and the iron has a +3 charge, thus net charge of -3. The referee is correct we do not experimentally determine the charge so our statement could have been misleading.

We have changed wording to

“The Fe^{3+} is coordinated in an octahedral manner by the six hydroxyls of the three catecholates which have been reported to be deprotonated at neutral pH⁹ resulting in a net charge for the molecule of -3.” We cite to J. Am. Chem. Soc. 1979, 101, 20, 6097-610

Line 113 Grammar

it simultaneously takes up the antibiotic into the periplasm;

Apologies, Corrected ‘Thus, as the bacterial cell transports the iron loaded siderophore that is essential to its survival it also takes up the antibiotic into the periplasm’

Line 115 results in patient with urinary tract infection and pyelonephritia²⁷.
In a patient or in patients?

Changed to : ‘in patients’, sorry

Line 139
with *E. coli* TBDT FepA which is also its closest structural relative with an r.m.s.d of 1.01 Å for 652 aligned Ca atoms. The *Pseudomonas* TBDT, PaPirA (72% homology and 60% identity) and the *Acinetobacter baumannii* homologue AbPirA (64% homology and 48% identity)

It would be useful to mention relevant pdb codes in this sentence.

We agree and have added the pdb codes as follows

PfeA has a 75% sequence homology (60% identity) with *E. coli* TBDT FepA (1FEP) which is also its closest structural relative with a root-mean-square deviation (r.m.s.d) of 1.01 Å for 652 aligned Cα atoms. The *Pseudomonas* TBDT, PaPirA (5fp2, 72% homology and 60% identity) and the *Acinetobacter baumannii* homologue AbPirA (5fr8, 64% homology and 48% identity)

Line 216 Grammar

Like for Fe³⁺-enterobactin, Fe³⁺-BCS and Fe³⁺-TCS coelute

Modified to:

“Fe³⁺-enterobactin, Fe³⁺-azotochelin and Fe³⁺-protochelin coelute with PfeA on a”

Line 236

Apart from the mutation itself and an additional water molecule in R480A, the mutant complexes were essentially identical except that the density for the Fe³⁺-enterobactin molecule was weaker.

Could this weakness be quantified and discussed in the text by using B-factors, occupancies, anomalous signals of the iron ions ?

B factors are indeed higher (shown in Table1 in the SOI) and we thank the reviewer for noting this.

Sentences have been modified.

“Apart from the mutation itself and an additional water molecule in R480A, the mutant complexes were essentially identical except that the B-factors of the Fe³⁺-enterobactin molecule were higher reflecting either increasing mobility or a partial occupancy of the ligand.”

“The weaker binding at the first site is consistent the higher B-factor of the Fe³⁺ enterobactin molecule observed in the complex crystal structures.”

L238 the parent protein, the R480-Q482A double
Should be ‘the parent protein, the R480A-Q482A double ‘
Yes, and has been modified

Line 258

Protein has been incubated with the Fe³⁺-siderophore before been loaded on a
The sentence has been modified as follow:

“Protein was incubated with the Fe³⁺-siderophore before being loaded on a”

Line 282

very strong correlations between the extracellular loops that form the binding site
More discussion is need here to describe what sort of correlations and what they mean in terms of the proposed mechanism

We apologise to the reviewer for the lack of clarity.

We have now explained the correlations we see are structural changes and these are reflected in the hydrogen bond network. We infer that these correlated changes in conformation mean that binding in the site results in a change in the TonB binding region. This is (in our mind) yet more evidence for the new site being real; it can affect the TonB motif. We have tried to make this clearer in the text.

Reviewer #2 (Remarks to the Author):

This study identifies an apparent binding motif for the siderophore enterochelin on external surface of the Pseudomonas FepA homologue PfeA. The solved crystal structure in the absence of enterochelin ligand differs from those of other crystallized TBDT proteins in that the internal portions of the beta barrel are obstructed by the folding of external loops, leaving a small hole. Crystals soaked with enterochelin take up the siderophore, with those structures then showing enterochelin bound to residues comprising edge features of this small hole. Mutations in these residues partially support the contention that this is a biologically relevant binding site.

The central issue is whether or not this is real. The authors raise this question (Lines 297-298)"whether this new binding site in the extracellular loops was indeed genuine or an experimental artefact". This is followed by (Lines 342-343) with "We speculate that, by soaking, we may have ‘trapped’ the PfeA protein in the crystal in such a way that the first binding site is formed..."

We agree with the reviewer this is indeed the central question. We felt that we had shown very strong experimental evidence for this. However, we fully respect peer review and have taken two approaches to improve the paper.

1 We make more clear what we see experimentally, what we infer from the data and what we speculate based on interpretation and reasoning. In doing so, we soften some of the language from definitive statements.

2 We have obtained new data these data significantly strengthen the evidence base for the conclusions in the revised paper.

So yes, it looks like it is trapped in a specific conformation - the authors note that the loops in the E. coli FepA structure are more disordered what they see and that the π - π stacking found to be important in vivo is not evident (lines 174-176)

The π - π stacking is to be found in the site adjacent to the plug, we have now made this clear in the manuscript. The referee is correct there is no π - π stacking in the new (1st) site. There is as we make clear potential for π - π stacking in the second site close to the plug.

The binding structure of the protein is not however 'trapped', the protein adopts this conformation in response to the presence of the ligand (we make this point below). The change is shown in Figure 2d.

I think the authors recognize this, but the way everything is described suggests that this is the normal conformation, ligand binds, and then is moved into the internal chamber where it can then associate with the cork domain. What is not mentioned here is that we have known for a long time that TBDT ligands do bind to regions in the outer loops -work in the Braun lab in the 1990s (J. Bact.174:3479 & 177:694) identified loop regions in FhuA required for phage and colicin M docking that could be competed for with the siderophore ferrichrome and the siderophore-conjugated antibiotic albomycin. The idea of surface-exposed binding domains that first capture free ligands to enhance interaction with a second domain that mediates an energy-dependent event is well established and widespread in molecular systems.

We have now cited Braun studies (we chose the more recent, but the referee is correct we should have cited originally)

The referee uses the phrase "captured" and this does nicely illustrate our central message, that this new site is indeed the first place the ligand is bound by the protein.

With our new data, we think the paper makes a convincing case that the site observed in the crystal structure is real and biologically important. We have been careful to use language such as 'supports' rather than proves.

In new data, we compare this paper to the BauA siderophore receptor. We show BauA does not have these extensive loops, has a single binding site and simple ITC isotherm. The high-resolution crystal structure shows that in BauA the binding site is located adjacent to the plug. Our argument is not that all siderophore receptors have two binding sites on the extracellular side of the protein, but that there is compelling experimental evidence that the enterobactin ones (PfeA, FepA etc) do. This is very important since, if we are right, enterobactin based trojan horse antibiotics have to be recognised first by this extracellular loop region before it moves to the site adjacent to the plug domain.

This study identifies some residues that are important, and can "catch" diffusing enterochelin, even when locked into a rigid crystal structure. That does not mean this is a conformation that exists in vivo. Mutations could disrupt that binding, and residue G342 was essential (as a glycine probably more about structure than as a site itself) - R480 and Q482 are better candidates for interaction - and as the authors note (line 316) the residue corresponding to R480 in FepA (misidentified as "K83") has been shown to be important for enterochelin binding. In summary, this is a crystal structure with surface exposed residues that have a role in binding enterochelin. That does not mean this is a structure that exists in vivo - there are many other potential structures in which those residues would be accessible.

When the referee says "catch" we have assumed they imply binding in a non-specific low affinity way. We believe our data strongly argues for specific binding (see below).

The referee makes the point that we do not know the arrangement of the loops that exists in the crystal is the same as in solution. This is of course true for all crystal structures, what the

structure does is guide further experimentation and a conformation in the crystal is generally assumed to be possible in solution (even if not the most favoured form).

When we add enterobactin (or two other different compounds) the loops in the structure adopt a new arrangement (Fig2D); this is evidence for a real binding site. We have made several mutants of these loops and crystallised these structures complexed to enterobactin. All of the functional mutants show the same structural re-arrangement of the loops upon enterobactin binding, whilst not proof the complex is “real” it is experimental evidence. As a consequence of the loop rearrangement the protein binding site makes extensive van der Waal interactions with enterobactin, seven specific hydrogen bonds to enterobactin and inserts residues into the grooves generated by the 3-fold symmetry of enterobactin. These are characteristics associated with specific recognition.

Further evidence for complementarity comes from molecular modelling which suggest the complex has strong binding (based on new data where we calculate the binding lifetime) and docking experiments which suggest an excellent fit. Forcing the enterobactin into the double mutant protein gives a complex with a 10,000 times shorter lifetime (new data).

In other new data we show the apo structure which has a different arrangement of the loops docks the ligand with a much a much lower free binding energy (Fig S9).

As the referee says, mutation of the residues that the structure shows bind to the ligand reduces binding affinity. We respectfully disagree with the referee’s emphasis, “some of the residues are important”, rather we would say that guided by the structure, we were able to reduce and eliminate binding by a series of targeted mutagenesis; thus identify essential elements of binding from the crystal structure. Co-complexes with mutants that exhibit reduced binding affinity show that the ligand is less well ordered (higher B-factor). The double mutant has a native structure (shown by a crystal structure) but does not bind the ligand *in vitro*, *in vivo* and does not take up enterobactin in cells. We argue that the structure has suggested a model, which we set out to test experimentally.

In revising the manuscript, we have tried to be clear where uncertainty lies, take note of the referee’s comments and soften our language. We do however think that the most parsimonious explanation of our data is that the protein has a binding site located in the extracellular loops.

As a related point, the studies with azotochelin and protochelin (Fig 4) do not address the question of whether or not the solved structure is relevant - they are simply *in silico* predictions consistent with the the above *in vitro* observation.

These are experimental crystal structures they are not *in silico* models, we apologise this was not clear in the original version. The fact that, despite the chemical differences in the ligands, they result in the same conformational change in the loops of PfeA in order to bind, we take as further evidence for the binding site being biologically meaningful.

The second major point involves the iron transport assays (Fig. 6). All of the appropriate controls are present, and so the focus is on the comparison of the plasmid-borne wild-type (pMMBpfeA) and the double mutant (pMMBpfeAR480AQ482A). The unresolved question here is "are the slopes different?" Baseline for the wild type is ~80 and it climbs to ~120 (why does it start so much higher than all the others? - was it more highly expressed so you have more loading prior to transport - did you check expression levels?); the double mutant baseline is around 25 but drops to 20 and then is at 50 and ends at 40. So the transport is a

little less than a doubling of ligand over the 30 minutes. Same is true for the wild type. Statistics would be useful here (what are the slopes and what are their r values? "...experiments have been repeated three times and equivalent kinetics have been observed" (line 567) is not helpful.

We have addressed this criticism with new experiments and changed our approach.

We have done the following

- (1) We measured the level of RNA coding for the PfeA protein in PAO1, the knockout cells with native protein and knockout cells with the double mutant PfeA. These show that the knockout strain with the native protein encoded by the plasmid has 160-fold higher level of RNA than wild type PAO1 (the double mutant has around 100-fold more protein). Error bars are given in the paper. These data clearly show that we have a transcription of the *pfeA* gene (wild type and mutant) carried by the pMMB plasmid.
- (2) We carried out a membrane preparation and analysed by SDS-PAGE, this shows that the knockout cells have much higher levels of expression of both the native and double mutant protein (in the outer membrane than in wild type PAO1). These results essentially confirm the RNA results above, but do establish both the native and double mutant PfeA protein are being made and directed to the outer membrane. (These findings echo *E. coli* heterologous expression for structural biology which showed the double mutant PfeA can be purified with similar yields to native).

Together these experiments establish that the knockout system has high levels of the native and the double mutant PfeA in the outer membrane when plasmid are added, thus the knockout system can be reliably used to measure PfeA function. Our structural works show the double mutant PfeA folds normally and the presence in the outer membrane is consistent with properly folded protein. We believe these new experiments answers the referee's very valid concerns about expression levels. We are grateful for this suggestion.

With this more reliable grounding, we approached the *in vivo* function in a more straightforward way using two related assays. We have removed the kinetics analysis used previously.

In first assay, we measured enterobactin binding only, this is done by pre-treating the cells with CCCP to shut down the energy system required for uptake.

These data show that knockout cells, with native PfeA encoded on the plasmid, binds far more iron than wild type PAO1. This would be expected given the higher level of native PfeA on the outer membrane of the knockout cell (RNA level and SDS PAGE). The observed binding also reconfirms the SDS-PAGE result that protein encoded by the plasmid is properly folded.

The significant comparison for our argument is between the double mutant PfeA on plasmid, native PfeA on plasmid and the control no plasmid. These show that the double mutant PfeA protein in knockout cells and the no plasmid knockout cells bind MUCH less iron than knockout cells with the native PfeA on the plasmid.

The no plasmid knockout binding shows there is background level of binding, since there are other TBDTs in *Pseudomonas* that bind enterobactin, not just PfeA, this background is to be expected. The crucial finding is that the no plasmid and double mutant PfeA protein show the same low (background) level within error. Thus, we conclude the presence of double mutant PfeA, even though expressed at a very high level, does not show any detectable increase of

enterobactin binding compared to the knockout cells without plasmid. The double mutant, we conclude, does not bind enterobactin *in vivo*, echoing the conclusion from the *in vitro* work.

In a second assay, the CCCP was removed allowing uptake of enterobactin to proceed. The no plasmid knockout cells show a small but measurable increase in iron. This is expected since the other TBDTs in the *Pseudomonas* cell, that bind enterobactin, will also now become active in uptake when CCCP is removed. This is the new background in active uptake cells. For knockout cells expressing the native protein we see a larger (than the no plasmid knockout cells) increase in the iron content of the cells. We conclude that the presence of native PfeA in the knock out strain increases the iron content of the knockout strain because enterobactin is being actively taken up by the native plasmid encoded PfeA protein. The crucial finding for our paper is that no plasmid and the double mutant PfeA give identical results (within error), the same smaller increase in iron. Our data show that despite, the presence of significant levels of the double mutant PfeA in the knockout cell outer membrane, these cells do not show any detectable increase ability to take up iron when compared to knockout cells with no plasmid. The double mutant PfeA, we conclude, is not competent for the uptake of enterobactin *in vivo*.

We recognise this second result can be argued to be redundant, since no binding to double mutant PfeA *in vivo* (established in the first assay) would of course mean no transport by double mutant PfeA *in vivo*. However, we think the assay strengthens the paper's experimental base by further confirming the double mutant PfeA is inactive *in vivo* and that the plasmid encoded native protein is biologically functional in the knockout cells.

In the revision we soften our language to avoid saying we have proven. However, we think the assay data taken together establish the double mutant is not functional *in vivo*. Since we know the double mutant is properly folded, highly expressed and inserted the outer membrane then this lack of function can be confidently be attributed to an inability to recognise enterobactin. Since the mutants were based on the understanding of interactions between PfeA and enterobactin derived from multiple crystal structure, we do think this is strong scientific evidence for our thesis.

Other aspects to address are:

1.)The is a general lack of qualification of statements. For example (line 334, line 362) "Molecular dynamics shows..." No it doesn't - models suggest - they do not show.

We have changed the language.

2.)There is a wide use of abbreviations and acronyms without definition.

We have corrected these.

3.) there are a number of typos and grammatical issues that can be readily cleaned up - but be careful with aspects that aren't evident, like the error on line 316 with the FepA residue (see above).

We apologise for these and have corrected them.

In closing, this is a good study, but over-interpreted..

R.A. Larsen

We thank Professor Larsen for the review and his complement "good study". We have revised the manuscript as detailed above to address his comments. We think the new data and

the more careful wording mean the manuscript is much stronger and avoids over interpretation.

Additional Figure for reviewers

We have uploaded for the reviewers a figure showing PAO1 versus the plasmid encoded PfeA uptake assay, we can put this in supporting if desired but we do not feel it is a relevant comparison that our paper relies on.

These data show that the knockout strain with native PfeA is less efficient (although functional) at transport than wild type PAO1.

This is interesting but not, we feel, relevant here, our paper is focussed on the comparison of the double mutant and native PfeA as a means to confirm the significance of the binding site identified by crystallography.

The knockout cells give clear data that

- (1) the native binds and transports enterobactin
- (2) the double mutant (within error) does not bind or transport enterobactin.

The issue of the biology of the knockout cells compared to PAO1 is not relevant to this question. Our interpretation of the difference between wild type PAO1 and knockout with PfeA; is that the very high (unnaturally so) levels of PfeA in the outer membrane may in part disrupt the TonB system (it is still functional). Doing the experimental work to dig out the effects on the outer membrane and the TonB system caused by overexpression of PfeA whilst important for other purposes, are beyond this paper.

Figure for referee only

⁵⁵Fe-enterobactin binding and uptake in PAO1 and $\Delta pfeA$ (pMMBpfeA). Cells were grown and were incubated with or without 200 μ M CCCP before initiation of transport assays by the addition of 500 nM ⁵⁵Fe-enterobactin. After 30 min incubations, the radioactivity accumulated in the bacteria was counted. The results are expressed as pmol of ⁵⁵Fe-enterobactin bound and transported per ml of cells at an OD₆₀₀ of 1. The experiments were repeated three times.

REVIEWERS' COMMENTS:

Reviewer #1 (Remarks to the Author):

The revised paper has satisfactorily addressed all the points I raised in reviewing the first version. The paper has also been improved by clarifying some of the ambiguous statements in the previous version and by replacing the poorly refined structure with a better defined higher resolution structure.

Reviewer #2 (Remarks to the Author):

The authors have done an excellent job addressing my concerns and, in my opinion, the concerns of reviewer one. This is a strong manuscript and will make a significant addition to the literature.

best regards,
Ray A. Larsen